January 21, 2018

Dear Reviewer and AMT Editorial Office,

The attached files are ready for production of the accepted manuscript.

Thank you for you the thoughtful reviews of this work, and for accepting it for publication in AMT.

Note that a URL directing readers to data in an online repository has been added.

Sincerely,

Dr. Caroline Alden and co-authors

# Methane leak detection and sizing over long distances using dual frequency comb laser spectroscopy and a bootstrap inversion technique

Caroline B. Alden[1,2*], Subhomoy Ghosh[3], Sean Coburn[1], Colm Sweeney[2,4], Anna Karion[3], Robert Wright[1], Ian Coddington[3], Kuldeep Prasad[3], Gregory B. Rieker[1]

[1]Department of Mechanical Engineering, University of Colorado at Boulder, Boulder, CO, 80309 USA
[2]Cooperative Institute for Research in Environmental Sciences, Boulder, CO 80309, USA
[3]National Institute of Standards and Technology (NIST), Gaithersburg, MD, 20899, USA
[4]National Oceanic & Atmospheric Administration (NOAA), Boulder, CO 80305, USA

*Correspondence to*: Caroline B. Alden (caroline.alden@colorado.edu)

**Abstract.** Advances in natural gas extraction technology have led to increased activity in the production and transport sectors in the United States, and, as a consequence, an increased need for reliable monitoring of methane leaks to the atmosphere. We present a statistical methodology in combination with an observing system for the detection and attribution of fugitive emissions of methane from distributed potential source location landscapes such as natural gas production sites. We measure long (>500 m), integrated open path concentrations of atmospheric methane using a dual frequency comb spectrometer and combine measurements with an atmospheric transport model to infer leak locations and strengths using a novel statistical method, the non-zero minimum bootstrap (NZMB). The new statistical method allows us to determine whether the empirical distribution of possible source strengths for a given location excludes zero. Using this information, we identify leaking source locations (i.e., natural gas wells) through rejection of the null hypothesis that the source is not leaking. The method is tested with a series of synthetic data inversions with varying measurement density and varying levels of model-data mismatch. It is also tested with field observations of 1) a non-leaking source location and 2) a source location where a controlled emission of 3.1 E-5 kg s$^{-1}$ of methane gas is released over a period of several hours. This series of synthetic data tests and outdoor field observations using a controlled methane release demonstrates the viability of the approach for the detection and sizing of very small leaks of methane across large distances (4+ km$^2$ in synthetic tests). The field tests demonstrate the ability to attribute small atmospheric enhancements of 17 ppb to the emitting source location against a background of combined atmospheric (e.g., background methane variability) and measurement uncertainty of 5 ppb (1-sigma), when measurements are averaged over 2 minutes. The results of the synthetic and field data testing show that the new observing system and statistical approach greatly decreases the incidence of false alarms (that is, wrongly identifying a well site to be leaking) compared with the same tests that don't use the NZMB approach, and therefore offers increased leak detection and sizing capabilities.

# 1 Introduction

The combustion of natural gas in high-efficiency power cycles is cleaner and produces less climate-warming carbon dioxide gas than the combustion of coal (Environmental Protection Agency, 2015), which has led to interest in natural gas as a cleaner alternative to coal for energy generation. Advances in natural gas extraction technology have led to a 35% increase in total natural gas production between 2005 and 2013 in the United States (U.S. Energy Information Administration, 2015). Production is expected to increase by 45% above 2013 levels by the year 2040 (U.S. Energy Information Administration, 2015). A caveat to the promise of natural gas as a lower climate impact energy source, however, is that leaks of methane during extraction and delivery can result in climate warming. Methane gas has high global warming potential (GWP): much higher, for example, than carbon dioxide ($CH_4$ has a GWP of 28 over 100 years, compared with $CO_2$, which has GWP of 1 by definition (Myhre et al., 2013)). Above a low threshold (estimated to be $\approx$3.2% by Alvarez et al. (2012)) leak rate from well to power plant, the near-term climate impacts of using natural gas for power generation become worse than coal (Alvarez et al., 2012; Hayhoe et al., 2002). Recent system-wide analysis suggests that natural gas sector leak rates are likely higher than inventory estimates (Brandt et al., 2014; Zavala-Araiza et al., 2015a). To achieve the lower climate impacts and greater economic benefits

of domestic natural gas production, it is important to find low cost methods to detect and reduce methane leakage (Alvarez et al., 2012).

The current industry practice for leak detection and repair (LDAR) is to perform infrequent (annual or less for most sites) "spot" checks for leaks, for example by visual inspection with an optical gas imaging (OGI) camera. However, recent work has shown that methane concentrations measured by OGI cameras can be drastically underestimated if conditions are not ideal, for example under conditions of lower temperature values or higher wind speeds, or if viewing distances are greater than 50 m (Ravikumar et al., 2016). Furthermore, spot check monitoring is inadequate for detection of leaks, given strong evidence for intermittency of leaks (Allen et al., 2013, 2015a; Mitchell et al., 2015; Subramanian et al., 2015). It has been observed that a small number of facilities leaking at very high rates – so-called "super-emitters" (Brandt et al., 2014; Frankenberg et al., 2016; Rella et al., 2015; Zavala-Araiza et al., 2015b) – can account for a majority of total emissions (Allen et al., 2013, 2015a, 2015b; Brandt et al., 2014). These characteristics underscore the importance of continuous monitoring for leaks over large areas. Field campaigns with sophisticated atmospheric sampling techniques provide valuable snapshots of the state of natural gas development facility leaks (e.g. Brantley et al., 2014; Karion et al., 2013), but it would be too costly to employ such measurement strategies for long-term continuous monitoring of most natural gas sector facilities.

We present and test an atmospheric measurement system coupled with a statistical inversion approach for detecting and quantifying emissions of methane. The statistical approach is focused on limiting the occurrence of false-positive leak detection. The measurement system used to test the statistical approach is composed of a long-range open-path laser situated in the center of a field of well sites, and a series of retroreflectors around the perimeter of the field to direct light back to a detector co-located with the laser. The concentration of trace gases along the open beam path (defined as the path between the spectrometer-detector system and a retroreflector) is determined from the species-specific absorption of light (Dobler et al., 2015; Flesch et al., 2004; Groth et al., 2015; Hashmonay et al., 1999; Levine et al., 2016). Many open-path absorption methods for determining species concentration have been demonstrated (Akagi et al., 2011; Dobler et al., 2015; Flesch et al., 2004; Jones et al., 2011; Nikodem et al., 2015; Wagner and Plusquellic, 2016; Wu et al., 2014). Here we use a dual frequency comb spectrometer (DCS): a unique broadband, high-resolution spectrometer that offers very high stability (low drift) and measurement reproducibility of the trace gas measurement, so that concentrations can be compared across different conditions and times (Coburn et al., n.d.). It was recently demonstrated that two separate dual frequency comb spectrometers stationed side-by-side and measuring the same 1-km outdoor path showed methane concentration agreement to 0.35% over a two-week period under ambient variations in temperature, pressure and stability (Waxman et al., 2017). In principle, the range of conditions under which two separate dual frequency comb spectrometers should be comparable is much wider than ambient conditions, because the concentration retrieval is largely dependent on the quality of absorption models (which are well-defined under most conditions experienced at Earth's surface). Previous work also demonstrates that this method of atmospheric trace gas measurement does not require regular or traditional calibration (Coburn et al., n.d.; Rieker et al., 2014; Truong et al., 2016;

Waxman et al., 2017). Laboratory and initial field measurements made with the dual frequency comb spectrometer indicate extremely high measurement precision (3 ppb or lower) over long (1 km one-way, or 2 km round-trip) pathlengths (Coburn et al., n.d.; Rieker et al., 2014; Truong et al., 2016; Waxman et al., 2017). The combination of low uncertainty and high stability enable new opportunities for detection and sizing of even very small emissions of methane (Coburn et al., n.d.). Furthermore,

the demonstration of sensitive methane measurements over kilometer-scale open paths allows for monitoring methane concentrations over large areas such as natural gas production, processing, and distribution sites. While frequency comb measurements have previously been made in laboratory settings, the recent work of (Coburn et al., n.d.) along with the new work shown here demonstrate the viability of dual frequency comb spectroscopy in real-world conditions.

We use the dual frequency comb measurements in a series of synthetic data and field data tests to demonstrate the utility of the observing system and a novel statistical method for accurately locating one or more point sources of methane within a large area (4+ km$^2$) using distributed measurements of methane concentrations and an atmospheric transport model. Previous studies have used Gaussian plume models with atmospheric measurements of wind conditions and constituent concentrations to detect sources (e.g., Hirst et al., 2004), and past studies have also shown the utility of open-path lasers for measuring across-plume

concentrations for use in the detection of emissions (Flesch et al., 1995; McBain and Desjardins, 2005). Here, we present a novel statistical technique applied to source detection and quantification - with the goal of minimizing false positive source identification. The source-attribution method used here is to apply a non-negative least-squares fitting technique to solve for methane flux at a series of potential source locations (e.g., pads, well heads or other components), given a set of atmospheric observations and knowledge of atmospheric transport (Leuning et al., 2008). The new statistical approach, called the non-zero

minimum bootstrap method (NZMB), uses a bootstrapping of model uncertainties to produce an empirical distribution of source strength for a given well site. Specifically, the empirical distribution is obtained by performing multiple atmospheric inversions (or estimates of surface fluxes using atmospheric data) using a set of resampled atmospheric measurements. The NZMB method establishes a criterion by which well sites or facilities are identified as having non-zero methane emissions based on examination of the minimum value of an ensemble of inversions. That is, a potential leak site is positively identified

as a source of methane to the atmosphere if the empirical cumulative distribution of likely source strengths (determined with a series of bootstrap operations) does not include a minimum threshold flux such as zero. Similarly, a facility is identified as not leaking if the empirical cumulative distribution of likely source strengths does include the minimum threshold flux (that is, the minimum value of all bootstrap operations is, for example, zero). By defining a specific null value for each potential leak, this approach reduces the incidence of false positive leak identification (the incorrect attribution of a methane source to

a non-leaking facility or well), compared with the same tests that do not use the NZMB method (the "non-bootstrap" approach). For comparison, we run the same series of tests with the non-bootstrap approach, which approximates emissions using a single non-negative least-squares fit.

Synthetic data tests are performed that assess the effects of increasing measurement density (4, 8, 16, 32, and 64 beams), and the effects of increasing model data mismatch (that is, combined uncertainty in the ability to simulate observations arising from measurement, transport and other sources). Field tests with atmospheric observation data are performed in a 3 km x 2.5 km field site located in north-central Colorado over the course of one day in January, 2017. The meteorological conditions (wind speed, wind direction, atmospheric stability) on this day are typical of wintertime and annual mean conditions measured near the field site (for example, compared with conditions at nearby weather station KCOLONGM30). Field measurements are made along a series of 3 beams extending from a spectrometer in the middle of the domain.

We define leak identification success as maximizing the incidences of leaks found, with a minimal occurrence of false positive source identification, enabling quick response to leaks and avoiding costly mobilization of repair teams due to false positive leak identification. The ability to correctly ascertain the absence of a leak is therefore of equal importance to the ability to find leaks for regulatory compliance applications of this method. With the above tests, we therefore seek to determine 1) whether methane point source emissions can be detected and sized under conditions of observational uncertainty (model-data mismatch) and background variation, 2) whether the absence of a leak can be ascertained in an outdoor field setting, 3) whether the NZMB method allows for leaks to be positively identified under scenarios of greater simulated model-data mismatch uncertainty, compared with the non-bootstrap method, and 4) whether a higher number of observations increases likelihood that the NZMB and non-bootstrap methods can positively identify leaks. The success of the synthetic and field data tests demonstrates the potential of this observing system for continuous monitoring applications, such as for natural gas facilities, and for providing emission source locations and their approximate strengths. The experiments here also demonstrate the potential for this technology to be used for other source estimation and monitoring applications, for example carbon sequestration.

## 2 Methods

### 2.1 Gaussian plume atmospheric transport model

In both the synthetic and real data tests, atmospheric transport is simulated using a Gaussian plume model, using Pasquill-Gifford parameterization of plume dispersion in the lateral and vertical directions (Green et al., 1980; Griffiths, 1994; Hanna et al., 1982). Micrometeorology in the boundary layer is a non-trivial source of uncertainty for characterization of atmospheric flow, and the Gaussian plume model represents a simplified representation of atmospheric transport and dispersion. It is used to characterize the mean state (or steady-state) of source-receptor relationships with a point source, as long as the transport time from source to receptor is comparable to the data averaging time (Gifford, 1976; Hirst et al., 2004). More sophisticated plume (e.g., AERMOD) or stochastic Lagrangian dispersion models (e.g., WindTrax) and stability parameterizations would be expected to provide more robust representations of the wind shear and inhomogeneities in turbulence in the atmospheric surface layer (Flesch et al., 1995; Perry et al., 1994; Wilson and Sawford, 1996). We select the simplified and low-

computational-cost plume model for assessment of the NZMB method as a baseline test, rather than implementing more advanced representations of transport. Future campaigns aimed at quantification of true emissions will benefit from an assessment of the drawbacks inherent in Gaussian plume model characterization of atmospheric transport, or use of a more sophisticated model, particularly for measurements made at short range.

For the synthetic data tests, the choice of transport model is largely trivial, given that the transport is considered "perfect". Field data is collected with a constant methane source to the atmosphere and a measurement averaging time that is comparable to the source-to-receptor travel time, such that the Gaussian plume model is a simplified but appropriate choice of transport model (Gifford, 1976; Hirst et al., 2004). Because the purpose of this study is to confirm or reject the basic methodology and

not to investigate the impacts of micrometeorological representation on flux estimation, we find the plume model to be sufficient as a baseline test (see Sect. 6).

Neglecting influence of background methane concentrations, Eq. (1) shows the relationship between fluxes and atmospheric concentrations (e.g., Leuning et al., 2008):

$$\mathbf{c} = \mathbf{x} * (\mathbf{c}/\mathbf{x})_{modeled} \, , \hspace{4cm} (1)$$

Where the $n \times 1$ vector $\mathbf{c}$ is the atmospheric concentration of the constituent of interest, and $n$ is the number of measurements. The vector $\mathbf{x}$ is $m \times 1$ sources of the constituent (flux units), where the size of $m$ is equal to the number of potential source flux

locations. Here, the vector of fluxes, $\mathbf{x}$, is the emission rate of methane from each potential source location. In the synthetic tests and field tests described here, multiple measurements are made on each beam, such that $n$ is always greater than $m$. The value $(\mathbf{c}/\mathbf{x})_{modeled}$ is the transport operator matrix describing the relationship between the point source emission and concentrations at observation points (spectrometer beams) under different meteorological conditions, derived using the Gaussian plume model, and commonly written as $\mathbf{H}$ (that convention will be followed here) (see Sect. 2.5.3 for details on

scaling from point source, to point concentration, to line-averaged concentration).

## 2.2 Dual Frequency Comb spectrometer for long-range open path methane detection

Dual frequency comb spectrometer measurements are made by transmitting light from the spectrometer through open air at a discrete set of wavelengths where methane absorbs light. The light is transmitted in the direction of a retroreflector, which can be placed 1+ km away (Coburn et al., n.d.; Rieker et al., 2014; Truong et al., 2016; Waxman et al., 2017). The retroreflector

directs light back toward a detector co-located with the spectrometer. The amount of light that is absorbed by methane yields a direct measurement of the average concentration of methane along the open path from spectrometer to retroreflector. The measurements presented here are part of the first campaign to measure atmospheric concentrations with a fielded dual frequency comb spectrometer (Coburn et al., n.d.). The temporal resolution of measurements is related to averaging time: as

averaging time increases, measurement precision increases, until such time that atmospheric $CH_4$ variability begins to erode measurement repeatability (see Sect. 4.1). The spatial resolution of the measurement depends on beam length, which is easily adjusted by moving retroreflectors closer to or further away from the spectrometer; and beam width, which scales with telescope diameter.

## 2.3 Flux estimation with Non-Negative Least-Squares fitting solution

We use the Non-Negative Least-Squares (NNLS) algorithm in Fortran-90 to solve for a flux rate (that is, the emission rate from each potential source location), given atmospheric observations (synthetic or real) and atmospheric transport influence functions (Lawson and Hanson, 1995). This algorithm iteratively solves for the best-fit $m \times 1$ vector of fluxes, $\mathbf{x}$ (see Sect. 2.1 for a description of $\mathbf{x}$), given an $n \times 1$ vector of data measurements, $\mathbf{y}$, and an $n \times m$ matrix of influence functions, $\mathbf{H}$. Given $\mathbf{H}$ and $\mathbf{y}$, the NNLS algorithm computes a vector $\mathbf{x}$ (methane emission rate at each well site) that solves the least squares problem:

$$\mathbf{Hx} = \mathbf{y}, \text{ subject to } \mathbf{x} >= 0 \tag{2}$$

Uncertainties in $\mathbf{x}$ and $\mathbf{y}$ are not included in the NNLS fit; model-data mismatch is used only in generation of the synthetic observations, and not as a control on the solution for $\mathbf{x}$. The NNLS algorithm returns the solution vector, $\mathbf{x}$, and also allows for the calculation of $\mathbf{Hx}$, an $n \times 1$ vector describing the expected atmospheric concentration given $\mathbf{H}$ and the solution for $\mathbf{x}$.

## 2.4 Non-Zero Minimum Bootstrap Analysis

The non-zero minimum bootstrap analysis, or NZMB, is a statistical test of the null hypothesis (*Hypothesis$_0$*) that the source strength at a given well site is equal to 0 kg s$^{-1}$. It is used here to estimate source strengths in both the synthetic and field data tests. Whereas bootstrapping methods and least-squares methods are not novel techniques, and have previously been applied to problems of source strength estimation, we develop the present methodology with the motivation to seek a solution for fluxes in which the incidence of false-positive source attribution is limited (Efron, 1979; Lawson and Hanson, 1995).

For each of *m* potential source locations, the null hypothesis (*Hypothesis$_0$*) is that there is no methane emission from that potential source location, and the alternative hypothesis (*Hypothesis$_1$*) is that there is a (non-zero) emission from that potential source location:

*Hypothesis$_0$* : $x_j = 0$  ($j = 1,\dots,m$)

*Hypothesis$_1$* : $x_j > 0$  ($j = 1,\dots,m$)

Given that model-data mismatch uncertainty is not zero (i.e., there is uncertainty in the exact relationship between atmospheric observations and surface fluxes due to transport, measurement and other uncertainties), it is not expected that the NNLS fit of $\mathbf{Hx}$ to $\mathbf{y}$ is exact, although the problem is overdetermined (that is, $n > m$). We therefore use the mismatch between $\mathbf{Hx}$ and $\mathbf{y}$ to create an empirical distribution function describing the confidence interval of the fit to the data, and to accept or reject the null hypothesis claim that we have enough evidence to claim that a particular source is not leaking. That is, the empirical fit to the data is used to quantify uncertainties associated with the model-data mismatch (including, for example, instrument and

measurement uncertainties, transport uncertainties and model uncertainties) rather than relying on a "bottom-up" estimation of those sources of uncertainty. We rely on the assumption that model-data mismatch uncertainty has an un-biased Gaussian distribution. Although biases in transport or other sources of uncertainty can exist, we suggest that investigation of that contingency is suited for future studies.

The method for employing the bootstrap analysis is as follows. We first solve for surface-to-atmosphere fluxes of $CH_4$, $\mathbf{x}$, using NNLS, as described in Sect. 2.3. Second, for each observation, $y_i$ ($i = 1,\ldots,n$), we calculate the residual values from the fit to the NNLS solution:

$$\mathbf{e}_i = y_i - \hat{y}_i \,, \tag{3}$$

where $\hat{y}_i$ ($i = 1,\ldots,n$) are the individual values in the vector $\mathbf{Hx}$. The values of $\hat{y}_i$ ($i = 1,\ldots,n$) are the "predicted" change in atmospheric methane given the NNLS solution for $\mathbf{x}$, or the change in atmospheric methane that is simulated by convolving the source-receptor matrix, $\mathbf{H}$, with $\mathbf{x}$.

The next step in the NZMB method is: for each observation, $y_i$ ($i = 1,\ldots,n$), we generate a new estimate of that observation by
using Eq. (3) to sample from the vector of the residuals of the fit to the atmospheric data, $\mathbf{e}$, (with replacement, meaning a given value can be sampled more than once), and adding that randomly selected $e_i$ value to the predicted observation value, $\hat{y}_i$, to create $y_{bi}$ (Efron, 1979). That is, for each observation vector, $\mathbf{y}$, we create a new vector, $\mathbf{y}_b$ (b denotes a bootstrapped value):

$$y_{bi} = \hat{y}_i + \mathbf{e}_{bi} \tag{4}$$

We perform this step 1000 times, resulting in 1000 vectors $\mathbf{y}_b$, or 1000 different sets of observations of the form $\{y_{bi},\ldots,y_{bn}\}$, where $y_{bi} = \hat{y}_i + \mathbf{e}_{bi}$.

For the field data, we apply a moving block bootstrap (Künsch, 1989) because residuals of observations made nearer together in time are more likely to be co-representative, whereas residuals of observations made further apart in time are likely to be
less representative due to changes in wind conditions and atmospheric stability. We calculate the autocorrelation in time of the residuals resulting from a single non-negative least-squares fit and use for the moving block window length a value two times the lag time at which the autocorrelation falls below the 95% confidence level. As there is no time dimension in the synthetic data case, we do not apply the moving block bootstrap to those cases.

Next, we use NNLS to solve for $\mathbf{x}$ for each of the 1000 resampled sets of observations, yielding 1000 individual solutions for $\mathbf{x}$. The final step in the NZMB method is to apply the non-zero-minimum criterion to the 1000 bootstrap solutions for each member of $\mathbf{x}$. For each possible source location, we find the minimum value from the 1000-member bootstrap analysis. The non-zero-minimum criterion states that if the minimum bootstrap value for a given well location is 0 kg s$^{-1}$, then the source location is classified as having a leak rate of 0 kg s$^{-1}$ (i.e., no leak). This criterion establishes, under the null hypothesis, whether

or not 0 (< 0 is not possible since a non-negative least-squares fit is used) is included in the domain of the empirical cumulative distribution function with non-zero mass, described by the 1000 solutions for each well site in **x**. If zero is included in this distribution, then the null hypothesis (**x** = 0) cannot be rejected. Conversely, if 0 is not included in the empirical cumulative distribution function for a given well site ($x_j$), then the null hypothesis can be rejected and it can be assumed that the well site

is leaking. We use a large number of bootstrap members (1000) to ensure that the law of large numbers (LLN) is met. LLN justifies that when the number of bootstrap operations is large, the bootstrapped leak mean approaches the estimated leak from the sample (i.e., the bootstrapped leak mean is a consistent estimator of the estimated leak), and the distribution of the bootstrapped leak approaches the probability distribution of the source strength. Thus, we can claim that the bootstrapped estimator is a good candidate of the estimated leak from the NNLS, and that the empirical cumulative distribution function is

an approximation of the true cumulative distribution function.

After having identified which source locations are non-zero sources to the atmosphere (leaking), the mean leak strength is estimated as the mean of the 1000 bootstrap solutions for that source location. Uncertainty in the strength of the true leak is calculated as the standard deviation of the 1000 bootstrap solutions at the true leak location.

This method requires little additional computational cost over the non-bootstrap NNLS approach, because additional runs of the transport model are not required, only additional NNLS fits using resampling of the observations. The NZMB approach has the benefit of reducing false positive solutions while also gathering information regarding the parameters of the assumed Gaussian distribution.

**2.5 Synthetic Data Tests and Results**

**2.5.1 "True" leak locations and strengths**

To prepare synthetic data testing of the NZMB method, we randomly distribute 20 possible leak source locations within a theoretical 2 km x 2 km domain. This is a reasonable approximation of well density based on high-production regions of the western United States (average well density across the Marcellus and Haynesville shale gas plays are 3+ wells km$^{-2}$). In the

synthetic tests, therefore, $m$ = 20. Of the 20 well sites in the domain, we simulate a scenario in which 2 source locations are leaking. The "true" leak rate at well site number 6 is 4.5E-5 kg s$^{-1}$ and the "true" leak rate at well site number 19 is 3.0E-5 kg s$^{-1}$. The remaining 18 well sites are assigned "true" leak rates of 0 kg s$^{-1}$ (Fig. 1). The two non-zero leak strengths are very small: roughly half the size of the smallest leaks found by Rella et al. (2015) in a survey of oil and natural gas well pads. The height above ground level of each leak is 1 m.

### 2.5.2 Idealized meteorological conditions for synthetic data tests

The meteorological data used for synthetic data tests includes many wind directions and a variety of wind speeds during the sampling of each beam in the domain, representing an ideal scenario for the generation of as many independent measurements of the leak strength as possible. Leak strengths are constant through time, such that the time dimension of the meteorology does not need to be considered. This approach assumes that enough time has passed for all meteorological conditions to have occurred during the sampling of each beam, a condition that eliminates complications in comparing synthetic cases with different beam orientations. The idealized meteorological field applies 216 unique wind conditions to all beams: three wind speeds (2 m s$^{-1}$, 3 m s$^{-1}$ and 6 m s$^{-1}$) from 72 directions (from 5º to 360º, in 5º increments). The conditions represent a situation in which, over a long period of time, many different wind conditions yield a variety of different measurements downwind of emissions. Given the simple beam configuration presented here, which is agnostic of potential source locations, increasing the number of measurement conditions improves the conditioning of the problem (Crenna et al., 2008; Flesch et al., 2009).

### 2.5.3 Measurement system configuration and synthetic observations

The "synthetic" atmospheric measurements are simulated based on the dual frequency-comb spectrometer observing system described in Sect. 2.2. The spectrometer is located in the center of the domain, at x = 1000 m and y = 1000 m (Fig. 2). Configurations of 4, 8, 16, 32, and 64 beams per spectrometer-detector system are tested. In all beam configurations, retroreflectors are placed at an equal distance (1000 m) from the spectrometer and at equal distances from neighboring retroreflectors (e.g., Fig. 2). The hub-and-spoke beam configuration is a simple and repeatable pattern for comparison of different numbers of beams. The height of the spectrometer and retroreflectors is 3 m above ground level. Figure 2 shows beams, beam end point locations (retroreflectors) and the spectrometer in a case with 16 beams.

The vector of "true" atmospheric methane concentrations, **c**, is simulated by combining knowledge of atmospheric transport with knowledge of "true" sources and measurement (beam segment) locations with Eq. (1). The influence functions describing the relationships between each element of **x** and each segment of each beam path for each wind condition, **H**, are created using the Gaussian plume model described in Sect. 2.1, with neutral stability conditions (Pasquill category D). In order to generate the synthetic measurement data, each beam path is discretized into 100 segments. For each unique wind condition, "true" source fluxes are multiplied by **H** to calculate atmospheric enhancements at each of the 100 points along the beam path. Enhancements due to leaks are calculated independently for each segment of a beam and subsequently averaged for each beam and for each wind condition. This value mimics the actual data output of the spectrometer, which measures the average concentration along the beam length.

The dimensions of *n* (e.g., the length of the atmospheric concentration vector, **c**) in the synthetic tests vary along with the number of beams per spectrometer-detector system and the number of meteorological conditions. In the configuration of 4

beams, for example, $n = 216 * 4$, because each distinct meteorological condition is applied to each beam. In the 8 beam configuration $n = 216 * 8$, in the 16 beam configuration $n = 216 * 16$, and so on.

**2.5.4 Perturbation of observations with noise equivalent to model-data mismatch uncertainty**

Model-data mismatch is the difference between the true atmospheric $CH_4$ concentration, **c**, and the simulated or measureable atmospheric $CH_4$ concentration. This difference is expected to be non-zero due, for example, to measurement uncertainty (sampling and instrumental error), transport uncertainty (imperfect knowledge of air flow between source and observation points), and representation error (for example, the assumption that the measured segment of beam appropriately characterizes the atmospheric concentration at the time and space scales that it represents in the model). We assume here that uncertainty due to the imperfectly known background concentration is also part of model-data mismatch uncertainty. We simulate progressively larger levels of model-data mismatch in order to identify differences in model capabilities to locate and size leaks between the NZMB and non-bootstrap methods.

A range of model-data mismatch values are tested with the expectation that both the NZMB and non-bootstrap models will be more likely to locate and source leaks when lower model-data mismatch is added to the data. To simulate different possible magnitudes of model-data mismatch, the simulated true atmospheric concentrations, **c**, are perturbed with random Gaussian noise with mean 0 ppb and standard deviation equal to the following values: 0.1, 0.2, 0.3, 0.4, 0.5, 1.0, 1.5, 2.0, 2.5, 3.0, 3.5, 4.0, 4.5, 5, 6, 7, 8, 9, and 10 ppb, over a 1 km path. Measurement statistical uncertainty alone is expected to be on the order of 3 ppb or lower for a 1 km path (Rieker et al., 2014). As the results of field tests will show, the range of model-data mismatch values tested are an appropriate approximation of observed uncertainty (Sect. 4.4). Model-data mismatch uncertainties are assumed to be uncorrelated, following convention and understanding of the dual frequency comb measurement scheme. In Eq. (5), $\boldsymbol{\epsilon}$ is a vector of model-data mismatch uncertainty corresponding to the vector, **c**. Both vectors are of length $n$ $(i = 1,\dots,n)$, where $n$ is the number of observations, as described in Sect. 2.5.3. The vector **y** contains the synthetic observations, or the true atmospheric concentrations perturbed with measurement noise.

$$\mathbf{y}_i = \mathbf{c}_i + \boldsymbol{\epsilon}_i \tag{5}$$

**2.6 Field Data Observations**

**2.6.1 Description of field deployed dual comb setup**

The first measurements from a field deployed dual frequency comb spectrometer are from the NOAA/ESRL Table Mountain Test Facility, 10 km north of Boulder, Colorado (Fig. 3) (Coburn et al., n.d.). The spectrometer is located near the center of a large ($\approx 3 \times 2.5$ km) flat-topped mesa that rises several meters above the surrounding terrain (see Fig. 3). The dual frequency comb is housed inside of a trailer, with telescope transceiver affixed to a rotating gimbal on the trailer roof (roughly 4 meters above ground level). The actual dual frequency comb spectrometer is contained in a 56 x 56 x 61-cm electronics rack, and the

large trailer provides a field deployment home base. The beam transceiver system sends light between 1620 and 1680 nm, with discrete line spacing of 0.002 nm, through a 2-inch telescope. Dual-comb spectroscopy uses a large spectral bandwidth and high spectral resolution, which allows for the simultaneous fitting of the absorption pattern for each gas, so that interference among gases is avoided. Background infrared light does not affect the laser signal due to the heterodyne nature of the detection – the detected beat signals between the comb teeth are of high frequency whereas background signals (for example from solar radiation) are of lower frequency. The system emits and senses approximately 28,900 individual comb teeth (Coburn et al., n.d.; Rieker et al., 2014). The wavelength "window" to which the instrument at Table Mountain is tuned is ≈50 nm, spanning 625 individual $CH_4$ features, 2,482 $CO_2$ features, and 133 $H_2O$ features. Intensity feedback, triggered data acquisition, and onboard phase correction are quasi-autonomous, enabling the system to operate continuously for any length of time (Coburn et al., n.d.; Truong et al., 2016; Waxman et al., 2017).

### 2.6.2 Leak location and strength

For the field experiments at Table Mountain, a cylinder of compressed methane gas is placed roughly 528 m away from the spectrometer (Fig. 3) with the gas outlet 1 meter above ground level. The methane cylinder is outfitted with a regulator and an Alicat mass flow controller (MC-20SLPM-D). The flow controller is set to release methane in a controlled flow of 3.1 E-5 kg $s^{-1}$ at source location 1, between 10:08 and 16:30 on January 26, 2017. The flow rate at source location 2 is set to 0.0 kg $s^{-1}$ through the duration of January 26, 2017. The controlled methane release point is roughly 0.43 cm in diameter, and the velocity of gas exiting the tubing is negligible.

The field tests are arranged so as to approximate the synthetic tests as closely as possible: to emulate the "perfect" background condition of the synthetic tests, the background methane concentration for each source location is measured directly by an upwind beam (Crenna et al., 2008; Flesch et al., 2009). Because the background is assumed to be unique for each source location, each inversion includes only that source location in its solution for fluxes. That is, one inversion is performed for source location 1, and a separate inversion is performed for fluxes at source location 2. The dimensions of *m* for each test are, therefore, equal to 1, and the dimensions of *n* for each test are equal to the number of measurements made downwind of the source location.

### 2.6.3 Retroreflector locations

Three corner-cube retroreflectors are located near source locations 1 and 2 at Table Mountain (see Fig. 3). At their nearest points, the lateral distances between beams 1 and 2 and source location 1 are 11 m, and 6 m, respectively. The minimum lateral distances between leak location 2 and beams 2 and 3 are 12 m and 8 m, respectively. The horizontal distance from the spectrometer to each retroreflector is 584 m, 585 m, and 588 m, respectively for retroreflectors 1, 2 and 3. All retroreflectors are positioned 1 meter above ground level.

**2.6.4 Meteorology at Table Mountain**

Wind speed and wind direction are measured directly with a 3D Sonic Anemometer (RM Young 81000 Ultrasonic 3D Anemometer with manufacturer-specified accuracy of ± 0.05 m s$^{-1}$) located mid-way between the spectrometer and the retroreflectors. It is possible that local wind circulation could lead to meteorological conditions that are not homogenous across the Table Mountain site, which could cause the mean winds measured at the anemometer to not perfectly represent those influencing the plume. Measurement of the entire wind field is not practical, however, so the point measurement is used to characterize meteorology across the site. The suitability of the Gaussian plume model for short-range simulations decreases under low speeds, so all data taken at wind speeds below 0.8 m s$^{-1}$ was removed from this analysis (the reliability of the Gaussian plume model erodes as wind speeds decrease below ≈1 m s$^{-1}$) (e.g., De Visscher, 2013).

**2.6.5 Measurements**

We test the bootstrap methodology using measurements taken over the course of one day in January 2017. We test the ability of the bootstrap approach to both disprove the null hypothesis (i.e., to correctly ascertain the presence of a non-zero methane emission) and to prove the null hypothesis (i.e., to correctly ascertain the absence of a leak), by gathering measurements along beam paths that bound: 1) source location 1, where methane is released in a controlled flow rate of 3.1 E-5 kg s$^{-1}$, and 2) source location 2, where no methane is released. Quasi-continuous (626 Hz) data acquisition occurs for 2 minutes on each beam. Time-averaging over 2 minutes is performed to maximize gains in measurement precision as well as to average across shorter time scale eddy mixing events. After a measurement is taken, less than 30 seconds elapse while the gimbal moves to focus the beam on the next retroreflector ("retro") in the measurement sequence. The measurement sequence for the time period of study on January 26, 2017 is: retro 1, retro 2, retro 1, retro 3, retro 1, retro 2, and so on. A fourth retroreflector is included in the measurement sequence (leading to a small time delay between measurements made on retro 3 and retro 2), but data from that beam is not analyzed here for simplicity.

In the field tests, the dimensions of *n* vary along with the number of measurements taken on the beams used in the fit for the methane emission rate vector, **x**. For the fit to the methane emission rate at source location 1, all data (that is, all 2-minute measurements) gathered on retroreflectors 1 and 2 are used. For the fit to the emission rate at source location 1, all data gathered on retroreflectors 2 and 3 are used. Upwind measurements are used to constrain background, and downwind measurements are used to determine source strength. The dimensions of *n* are therefore equal to the number of downwind measurements. For the test at source location 1, *n* = 63 and for the test at source location 2, *n* = 30. The value of *n* is smaller at source location 2 because of the sampling pattern described above.

**2.6.6 Background CH$_4$ Estimation**

To most closely approximate the synthetic data testing framework in the field environment, we directly sample background CH$_4$ concentrations upwind of the leak point. The array of beams shown in Fig. 3 "sandwich" each source location. This configuration means that under most wind conditions (wind directions within $\approx 40^\circ$ of orthogonal to the beam array in either

direction), one beam is situated upwind and one beam is situated downwind of each source location. With this method, we attempt to remove the time-varying CH$_4$ concentration to which enhancements from discrete near-field emissions are added. While the Table Mountain site is relatively removed from expected anthropogenic and biogenic methane sources, the presence of nearby small livestock and oil and gas operations means that the background methane concentration does vary according to wind direction and through time. The "beam sandwich" approach, of placing beams on either side of each source location,

represents a plausible solution to future regional-scale monitoring of many potential emitters.

**3 Results of Synthetic Data Tests**

**3.1 Synthetic source location with and without the NZMB method**

We calculate solutions for **x** using NNLS in a single solution without a bootstrap approach for each set of beam configurations and for each model-data mismatch scenario in the synthetic data case. Figure 4 summarizes the findings of each test by

categorizing the results into four outcomes: 2 true leaks found with no false positives, 1 true leak found with no false positives, 0 true leaks found with no false positives, and 1 or more true leaks found with 1 or more false positive. The top half of Fig. 4 (for the non-bootstrap method) shows that, of the 5 different beam configurations tested, all result in false positive source locations under every model-data mismatch scenario when a non-bootstrap approach is taken. That is, even with very low model-data mismatch (0.1 ppb) and many beam measurement locations (64), the non-bootstrap method fails to positively

identify true leak sources without also generating false positive results. Non-zero solutions are found for source locations where no "true" leak exists.

The bottom half of Fig. 4 shows the results of the same tests, using instead the NZMB method for locating leaks. The results show that success in leak detection is much higher using NZMB, compared with the non-bootstrap tests. Indeed, none of the

NZMB tests result in the occurrence of a false-positive leak location, and only tests with low numbers of beams relative to the number of source locations (4 and 8 beam cases) fail to find both of the true leaks. The 4-beam case results in positive identification of both leaks up to a model-data mismatch threshold of 2 ppb, above which 1 true leak is found. One leak is consistently found up to a threshold of 5 ppb, and above 5 ppb model data mismatch no true leaks are identified (but no false positives are generated either). The 8-beam case results in accurate location of both true leaks up to a model-data mismatch

threshold of 3.5 ppb, above which 1 true leak is found (with no false positives). One leak is consistently found up to the maximum testing point of 10 ppb. In order to reliably locate both true leaks with no false positive results under all model-data

mismatch scenarios, 16 or more beams are needed for the set of cases that are tested here. Alternate configurations of "true" leaks at well sites other than 6 and 9 are not tested, however given that meteorological conditions are simulated equally from all directions, we would not expect a different set of results from a different set of "true" leaks.

5 A subset of the results for the 8-beam NNLS without bootstrap and the NNLS with NZMB cases are shown in Figs. 5 and 6 (for conciseness; all results are shown in the Supporting Information). It is evident from Fig. 5 that, even with very low model-data mismatch noise (0.5 ppb), the non-bootstrap model results in well sites other than the 2 true leak locations being erroneously identified as sources of methane. It is evident from Fig. 5 that, as model-data mismatch increases, the strength of incident false positive results also increases. By contrast, no false-positive leaks are identified in the NZMB case shown in

10 Fig. 6, at any level of model-data mismatch noise. Above a model-data mismatch threshold of 4 ppb, only one of two true leaks is found in the 8-beam case using NSMB. As Fig. 4 shows, 16 or more beams are necessary to consistently find both true leaks at higher thresholds of model-data mismatch uncertainty using the NZMB method, given the hub-and-spoke beam placement scheme tested here. More complex placement of beams (for example placing beams closer to known well sites) would likely result in even better ability to locate leaks with fewer beams.

15 **3.2 Synthetic source sizing using the NZMB method**

Synthetic data tests of the new bootstrap methodology presented here show high success in leak location, with zero incidence of false positive leak detections. Figure 6 shows the maximum and minimum values of 1000 bootstrap operations for each model-data mismatch test case for the 8-beam configuration. At low levels of model-data mismatch uncertainty (0.1-0.5 ppb), the maximum and minimum solutions bound a small range that is close to the true leak strength. As higher levels of model-

20 data mismatch noise are added to observations, the maximum and minimum values diverge. However, even as the maximum and minimum solutions diverge, most cases include the true leak strength within the maximum and minimum bounds.

Using the NZMB method, all beam cases (even the 4-beam case) correctly identify that both well sites 6 and 19 are emitting methane when model-data mismatch is 2 ppb or lower (Fig. 4). At that level of model-data mismatch, higher numbers of beams

25 and observations tend to lead to lower standard deviation around the mean estimated leak strength and a more accurate estimate of true leak strength (Table 1). An exception is at well site 19, where the 8-beam case did not perform as well as the 4-beam case. It may be that both cases were inadequate for accurately sizing leaks, and that 16 beams are necessary in a dense field of wells such as is tested here. The failure of the 8-beam case to accurately predict the leak rate at well site 19 is also evident from histograms of bootstrap operations, shown for each beam case with model-data mismatch of 2 ppb in Fig. 7.

Histograms of the results for the 16, 32 and 64 beam cases with 10 ppb model-data mismatch are shown in Fig. 8. It is clear from Fig. 8 that, even with very high model-data mismatch uncertainty, simple hub-and-spoke configurations of between 16-64 beams are able to locate and estimate leak flow rates to within reasonable bounds of uncertainty.

## 4 Results of Field Data Tests

### 4.1 Performance overview of field deployed DCS

Atmospheric observations are made over the course of one day on January 26, 2017 at the Table Mountain site. A set of 3 retroreflectors create long-range open-path beams of ≈585 m (Fig 3). Spectrometer performance in the field demonstrates no
loss of precision or reliability compared with laboratory performance, as demonstrated by (Coburn et al., n.d.). Figure 9 shows a plot of Allan deviations for January 26, 2017, demonstrating measurement precision of 5-6 ppb when measurements are averaged for 2 minutes. Precision of field measurements is limited by repeatability of measurements and atmospheric variability of $CH_4$ because measurements are time-averaged; the latter is likely a dominant driver of uncertainty in this case, as will be discussed in Sect. 4.3. The Allan deviation in Fig. 9 shows improvement of precision with averaging time, to a
minimum at ≈70 seconds, followed by an increase that is likely due to atmospheric variability.

### 4.2 Atmospheric Observations of $CH_4$ at Table Mountain

On January 26, 2017, measurements are made throughout the day, including during a 6.5 hour controlled release of methane at source location 1. At adjacent source location 2, no methane release is emitted. A series of 3 retroreflectors are oriented such that each source region is monitored independently from the other; one beam on either side of each source location serves as
a "background" measurement. We examine the results of two separate inversion tests: 1) a day-long set of observations of source location 1 (with the controlled release) that is situated between retroreflectors 1 and 2, and 2) a day-long set of observations of non-leaking source location 2 that is situated between retroreflectors 2 and 3. These tests are performed simultaneously, such that contamination from source location 1 could result in background contamination for monitoring of source location 2.

On January 26, 2017, mean wind speeds are 2.1 m s$^{-1}$ and winds are primarily from the east and northeast, so that retroreflector 1 is downwind of the controlled release, and retroreflector 2 is upwind of the controlled release. Similarly, retroreflector 2 is downwind of non-leaking source location 2 and retroreflector 3 is upwind of non-leaking source 2 (Fig. 3). Stability Classes range from B (moderately unstable) to D (neutral) throughout the course of the day (see Supplemental Information for a
timeseries of stability and detailed description of its calculation). We use the Griffiths (1994) corrections to the Briggs (1974) parameterizations to calculate $\sigma_y$ and $\sigma_z$.

At source location 1 (Fig. 10, panel a), during the period when the controlled release is on (non-zero flow), the downwind retroreflector (Retro 1) shows a clear enhancement above the concentration measured on the upwind retroreflector (Retro 2),
except during the middle of the day when the winds shift briefly to the south (Fig. 10, panel c). The mean of all $CH_4$ measurements along beam 1 during the period that the leak is on is 2046 ppb; the mean $CH_4$ measured along beam 2 during the same period is 2025 ppb. Both retroreflectors demonstrate changes in background $CH_4$ concentrations over the course of

the day; the range in values measured on the upwind retroreflector is 65 ppb. There may be a relationship between ambient $CH_4$ concentration and wind direction, as both retroreflectors show a drastic decrease in concentration when the winds abruptly shift to the West at 16:30 (which happens to coincide with the time the leak was turned off).

At source location 2, no leak is released during the period of study, and throughout the course of the day, both retroreflectors 2 and 3 measure similar changes in atmospheric $CH_4$ variability (Fig. 10 panel b). The range of measured values over the course of the entire day are 128 ppb on beam 2 and 124 ppb on beam 3. The mean of all $CH_4$ measurements (throughout the course of the day) is 2016 ppb on beam 2, and 2019 ppb on beam 3.

### 4.3 Background $CH_4$ Observations

The beams stationed upwind of each source location provide estimates of the background $CH_4$ concentration inflow for that location. After a linear interpolation to upwind measurements has been applied, this background is subtracted from measurements on the downwind beam, to yield a measure of the $CH_4$ enhancement due to fluxes at the source location. Applying this method, the mean and standard deviation of the enhancement above background on retroreflector 1 – which is downwind of source location 1 (leak rate of 3.1E-5 kg s$^{-1}$) – is 17.4 ± 10.1 ppb. Applying this method to source location 2, we

find the mean and standard deviation of the enhancement on retroreflector 2 – which is downwind of source location 2 (leak rate of 0 kg s$^{-1}$) – is 3.1 ± 7.3 ppb, a value within the range of variability expected from combined measurement and background uncertainty.

### 4.4 Field-based estimates of model-data mismatch

We examine measurements at source location 2, where no leak is present, in order to estimate model-data mismatch in the

field, for comparison with the model-data mismatch values applied in the synthetic data tests. By examining the difference between measurements made on different retroreflectors (retroreflectors 2 and 3) at similar points in time (within 5 minutes), we obtain an approximation of the combined contributions to model-data mismatch arising from measurement uncertainty, representation uncertainty, background construction (the method of background estimation) and background sampling (the method of sampling background concentrations). We find a standard deviation of 5 ppb. This value differs from the standard

deviation of the enhancement for the entire timeseries (reported above in Sect. 4.3) because it compares differences in upwind and downwind concentrations measured at approximately the same time. For estimation of total model data mismatch, we add (in quadrature) an estimate of the transport uncertainty that includes uncertainties in measurement of wind speed and wind direction, atmospheric stability parameterization, and placement of the sonic anemometer relative to the leak location (see Supplemental Information for detail). Transport uncertainty estimation is for a plume that interacts with any location along the

beam, and therefore requires knowledge of the mean distance between the leak point and each segment of the beam. The estimated transport uncertainty, calculated in this way, is 0.8 ppb. If, for example, the wind direction is perfectly perpendicular to the beam for the entirety of the measurement period (which does not occur on January 26[th], 2017), then the leak-to-beam

distance used in the calculation should collapse to the minimum lateral distance between the leak and the beam. Using that value instead, transport uncertainty is 12.2 ppb. The overall value of model-data mismatch (reflecting combined measurement, background, and transport uncertainty), estimated in this way, is therefore 5.1 ppb with a maximum range of 13.2 ppb, which suggests that the range of model-data mismatch values tested in the synthetic data experiments are appropriate. The Allan deviation in Fig. 9 shows a similar level of measurement uncertainty, suggesting that most of the uncertainty observed in our record is captured in this estimate of model-data mismatch, which includes effects of atmospheric variability. Precision could be improved by averaging data over a shorter time span (70 seconds), but those gains would be minimal (Fig. 9).

### 4.5 Results of inversions using Table Mountain observations

Both the Non-bootstrap and the NZMB approaches accurately predict the presence of methane emissions at source location 1 (Table 2). The average bootstrapped flux value is within 2-sigma of the true flux value measured at the flow meter at source location 1 (Fig. 11). At source location 2, the non-bootstrap approach falsely predicts a positive emission rate of 0.5 E-5 kg s$^{-1}$ (Table 2) where no leak is present. The NZMB approach, by contrast, is able to accurately predict that there is no leak present at source location 2, because the minimum of the 1000 bootstrap solutions is zero (Fig. 11). As the synthetic data tests also demonstrate, the NZMB method is necessary to avoid false identification of leaking source locations. The field data tests corroborate that the new bootstrap approach enables higher confidence of accurate attribution of emissions to source locations without generating "false alarms".

### 5 Discussion

The results of this study demonstrate success of the new observing system in finding one or more leaks of methane in a field of wells, using synthetic and field data for confirmation. The methods presented here for locating and sizing leaks of methane in a field of natural gas production facilities succeeds not only in identifying the location of a leak, but it also does so with no incidences of "false positive" leak detection in either the synthetic or field data tests.

### 5.1 Synthetic Data Tests

The results of the synthetic data tests demonstrate how the observing system tested in the field for a single source location can be expanded for simultaneous monitoring of many source locations. We find that synthetic tests performed without the NZMB methodology failed to identify the presence of leaks as reliably as synthetic tests performed with the NZMB method, demonstrating the improved robustness of this new statistical method for leak detection. In the non-bootstrap tests, all synthetic data cases resulted in false positive solutions (Fig. 4). By contrast, the NZMB method succeeds in correctly identifying two leaks of strength 3.0E-5 and 4.5E-5 kg s$^{-1}$ with 4 or more beams monitoring 20 wells in a 4 km$^2$ area, with 2 ppb model-data mismatch uncertainty (a condition that could conceivably also be met in the field given low background uncertainty and high measurement precision). The NZMB method also consistently succeeds in finding both leaks with 16 or more beams with at

least 10 ppb model-data mismatch uncertainty. Notably, the NZMB method locates and sizes both leaks with no false positive results. Determination of leak strength was successful to within 25% (and all but a few cases well below 10%) for all cases with 16 or more beams, using the NZMB method.

### 5.2 Field Data Tests

Field data testing of the NZMB method corroborates the synthetic data findings: that the new atmospheric observing system presented here results in high accuracy of leak detection without false positive results. The ability of the dual frequency comb spectrometer to identify a very small leak (3.1 E-5 kg s$^{-1}$), relying on very small methane enhancements (17 ppb) against a highly variable background (range of 65 ppb), demonstrates the potential utility of this method for methane leak detection over large areas.

An important caveat to the methodology presented here is the short length of the measurement averaging time, which presents a mismatch with the ideal application of most dispersion models (for which practice is generally to use averaging times longer than 2 minutes). This requirement in our methodology is due to two factors: the first is that rapid scanning for potential leaks is an important feature in areas where many sites must be monitored and leaks can be intermittent. The second factor is that

background methane concentrations can vary with high frequency (order minutes) in proximity to areas of oil and natural gas production (Dlugokencky et al., 1995). We attempt to mitigate uncertainties arising from using dispersion parameters developed for longer time-scale modeling over a 2-minute period in several ways. First, $n$ 2-minute dispersion calculations gathered over longer time scales ($n$ is between 30 and 63 for field data tests shown here) are aggregated for use in a single inversion, which is accepted practice (Scire et al., 2000). Second, both sources and receptors are close to the surface, which

may help to mitigate crosswind-integrated concentration fluctuation intensity (Weil et al., 2012). Third, a sensitivity test in which we adjust the horizontal dispersion coefficient, $\sigma_y$, for shorter time averaging, using the methods of (Gifford, 1976), shows negligible changes in the results (Supplemental Information). We find that the potential value of a method for rapid detection of methane emissions over large scales and against a highly variable background means that the uncertainties introduced from modeled eddy diffusivity parameterization are a complicating but not irreconcilable caveat.

### 6 Conclusions

The focus of this study is to show the powerful potential of the combination of a new statistical method with dual frequency comb spectroscopy for the location and sizing of point source emissions. The synthetic and field tests presented here rely on near-perfect (in the synthetic data tests) or well-constrained (in the field data tests) background concentration estimation. Future studies are needed to address the potential complications of more complex background conditions and meteorological

conditions under which it is not possible to obtain sequential "upwind" and "downwind" samples. Similarly, the tests here rely on the assumption of constant leak rates, which may not be a realistic assumption that can be made for methane emissions

from oil and gas operations. Future work to address these complexities will be necessary. Future studies are also needed to examine the gains that can be made from optimization of beam configurations for improved leak detection given variable wind and background conditions. In particular, previous work has shown the critical impact that sensor placement can have on the conditioning of the source-receptor relationship matrix (**H**), and suggests paths forward for optimization of sensor placement

(Crenna et al., 2008; Flesch et al., 2009). Specifically, the placement of one beam or sensor between each source to be apportioned would be expected to lead to a lower condition number and therefore a more reliable result (Crenna et al., 2008; Flesch et al., 2009). Work aimed at addressing these complications is underway, as are inversion efforts to resolve issues of leak intermittency.

A notable aspect of the micrometeorological modeling used here to demonstrate the NZMB methodology is the simple representation of atmospheric transport (the Gaussian plume model). The choice to use a simple model that is familiar to the broader scientific community is intentional, however its use belies the complex nature of turbulent mixing and dispersion in the atmospheric surface layer. What is gained in simplicity and in providing a baseline for the most basic performance of the methodology in a field setting may come at the cost of recommending a model that may not ultimately be well-suited for such

an endeavor. The Gaussian plume model neglects important aspects of atmospheric mixing such as wind shear and the height dependence of eddy diffusivity, and better models exist for simulation of atmospheric flow at this scale. It is assumed that more sophisticated models of atmospheric dispersion could, therefore, lead to better flux estimation. We suggest that future applications in field settings of the methodology presented here consider their use. Importantly, despite its drawbacks, the Gaussian plume model proves sufficient in the tests here for the accurate identification (and, importantly, avoidance of

misidentification) of controlled, field-based methane leaks. Future studies of the best transport model for the application of DCS measurements and the NZMB method for leak detection is warranted.

The initial work presented here demonstrates the promising potential of dual frequency comb spectroscopy for detection of leaks in the natural gas supply chain, and the valuable gains that can be provided by using the NZMB method over the NNLS

fitting technique alone.

The authors declare that they have no conflict of interest.

**Author Contribution**

S. Ghosh, K. Prasad, and C. Alden implemented the statistical NZMB technique. C. Alden, S. Coburn, R. Wright, C. Sweeney,

K. Prasad, S. Ghosh and G. Rieker designed the experiments and S. Coburn, R. Wright, and C. Alden carried them out. A. Karion provided expert guidance and experimental design input. C. Alden prepared the manuscript with contributions from all co-authors.

**Data Availability**

Data accessible at the following URL: ftp://aftp.cmdl.noaa.gov/user/alden/Alden_AMT_2018/

**Acknowledgements**

The information, data, or work presented herein was funded in part by the Advanced Research Projects Agency-Energy
(ARPA-E), U.S. Department of Energy, under Award Number DE-AR0000539. The views and opinions of authors expressed
herein do not necessarily state or reflect those of the United States Government or any agency thereof.

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

| Number of Beams | Well Site 6 Mean Strength | Leak One 1 s. d. | Well Site 19 Mean Strength | Leak Two 1 s. d. |
|---|---|---|---|---|
| 4 | 4.2E-5 kg s$^{-1}$ | 0.4E-5 kg s$^{-1}$ | 2.5E-5 kg s$^{-1}$ | 0.6E-5 kg s$^{-1}$ |
| 8 | 4.5E-5 kg s$^{-1}$ | 0.4E-6 kg s$^{-1}$ | 2.0E-5 kg s$^{-1}$ | 0.3E-5 kg s$^{-1}$ |
| 16 | 4.5E-5 kg s$^{-1}$ | 0.4E-6 kg s$^{-1}$ | 2.8E-5 kg s$^{-1}$ | 0.9E-6 kg s$^{-1}$ |
| 32 | 4.4E-5 kg s$^{-1}$ | 0.3E-6 kg s$^{-1}$ | 3.0E-5 kg s$^{-1}$ | 0.8E-6 kg s$^{-1}$ |
| 64 | 4.5E-5 kg s$^{-1}$ | 0.3E-6 kg s$^{-1}$ | 3.0E-5 kg s$^{-1}$ | 0.6E-6 kg s$^{-1}$ |
| | **True Leak: 4.5E-5** kg s$^{-1}$ | | **True Leak: 3.0E-5** kg s$^{-1}$ | |

**Table 1: NZMB Solutions for leak strength of true leaks, given 2 ppb model-data mismatch uncertainty, for each beam configuration.**

|  | Source Location 1 | Source Location 2 |
|---|---|---|
| **Controlled Leak Time On:** | 10:08 | NA |
| **Controlled Leak Time Off:** | 16:30 | NA |
| **Measured Mean Flow Rate:** | 3.1 E-5 ± 0.01E-5 kg s$^{-1}$ | 0.0 ± 0.0 kg s$^{-1}$ |
| **Non-Bootstrap Solution:** | 2.4 E-5 kg s$^{-1}$ | 0.5 E-5 kg s$^{-1}$ |
| **NZMB Solution:** | 2.6 E-5 ± 0.5 E-5 k/s | 0.0 ± 0.0 kg s$^{-1}$ |

**Table 2: Controlled methane release flow rates and 1 standard deviation for each field experiment, including local time that leak was turned on and off.**

**Figure Captions**

**Figure 1: Synthetic test observation area: 2 km x 2 km domain with 20 source locations (black dots) at randomly distributed x and y locations (position shown on x and y axes). Of 20 point sources, well site 6 (x=750, y=750) and well site 19 (x=650, y=1750) have non-zero source strengths (shown on the z-axis).**

**Figure 2: Map view of synthetic tests, with 20 source locations shown as black dots and 16 beams shown as gray lines that extend from the spectrometer (circle at x = 1000 m and y = 1000 m) to retroreflectors (black triangles).**

**Figure 3: Map view of observation test site at Table Mountain, Colorado (upper left inset shows geographic location of test site), with two source locations (location 1, in red, between beams 1 and 2; location 2, in green, between beams 2 and 3) and three beams shown as white lines that extend from the spectrometer (blue square) to retroreflectors (white triangles, and labeled 1-3).**

**Figure 4: Summary of synthetic data test results. Top 5 rows show results of non-bootstrap inversions and bottom 5 rows show results of NZMB inversions for the 4, 8, 16, 32, and 64 beam cases. Columns indicate results for different values of model-data mismatch added as noise to the synthetic measurements. Color coding of cells indicates summary of model success, as detailed by the legend.**

**Figure 5: Top left panel shows well site numbers (x-axis) and corresponding "true" leak rates (y-axis), and remaining panels show**
**resulting leak rate (y-axis) at each well site (x-axis) from non-bootstrap least squares fit to synthetic observations perturbed with model-data mismatch (MDM) noise shown, for the 8-beam case. Open circles show locations and strengths of all non-zero solutions.**

**Figure 6: Top left panel shows well site numbers (x-axis) and corresponding "true" leak rates (y-axis), and remaining panels show NZMB results (y-axis) for each well site location (x-axis) with synthetic observations perturbed with model-data mismatch (MDM) noise shown, for the 8-beam case. Light gray (black) open circles show locations and strengths of the maximum (minimum) of 1000**
**bootstrap operations. Minimum values of zero are not plotted.**

**Figure 7: Histograms of source strength, with mean ± 1 standard deviation shown with vertical lines for well site 6 (black) and well site 19 (gray), for each beam configuration, and with 2 ppb model-data mismatch uncertainty. Note that x-axes are truncated at 2E-5 kg s$^{-1}$ (lower bound) and 5E-5 kg s$^{-1}$ (lower bound) for scale.**

**Figure 8: Histograms of source strength, with mean ± 1 standard deviation shown with vertical lines for well site 6 (black) and well**
**site 19 (gray), for 16, 32 and 64 beam configurations, and with 10 ppb model-data mismatch uncertainty.**

**Figure 9: Allan deviation plot showing changes in measurement precision with averaging time from field data collected at Table Mountain on January 26$^{th}$, 2017.**

**Figure 10: Line integrated atmospheric CH$_4$ concentrations measured on January 26, 2017 along beam paths to retroreflectors 1 and 2 (a), and to retroreflectors 2 and 3 (b), and wind speed and wind direction (c). Gray and black points and left-hand axes of**
**panels (a) and (b) show CH$_4$ concentration. The black line and right-hand axis in panel (a) shows the flow rate at source location 1 (bounded by retroreflectors 1 and 2) and the black line and right-hand axis in panel b shows the flow rate at source location 2 (bounded by retroreflectors 2 and 3). In panel (c), the black line and left-hand axis show wind speed and the gray diamonds and right-hand axis show wind direction (according to meteorological convention, 0$^o$ = north, 90$^o$ = east, 180$^o$ = south, 270$^o$ = west, and 360$^o$ = north). All data reflects 2-minute averaging time.**

**Figure 11: Histogram of NZMB estimated source strength at source location 1, with dashed line showing the bootstrap mean and thin dotted lines showing ± 2 standard deviation. The thick black line shows the true leak strength at source location 1 (3.1 E-5 kg s$^{-1}$).**

**Figure 12: Histogram of NZMB estimated source strength at source location 2, with dashed line showing the bootstrap mean and thin dotted lines showing ± 2 standard deviation. The thick black line shows the true leak strength at source location 2 (0 kg s$^{-1}$). The**
**presence of 0 kg s$^{-1}$ in the histogram triggers acceptance of the null hypothesis (that the emissions rate at this site is zero).**

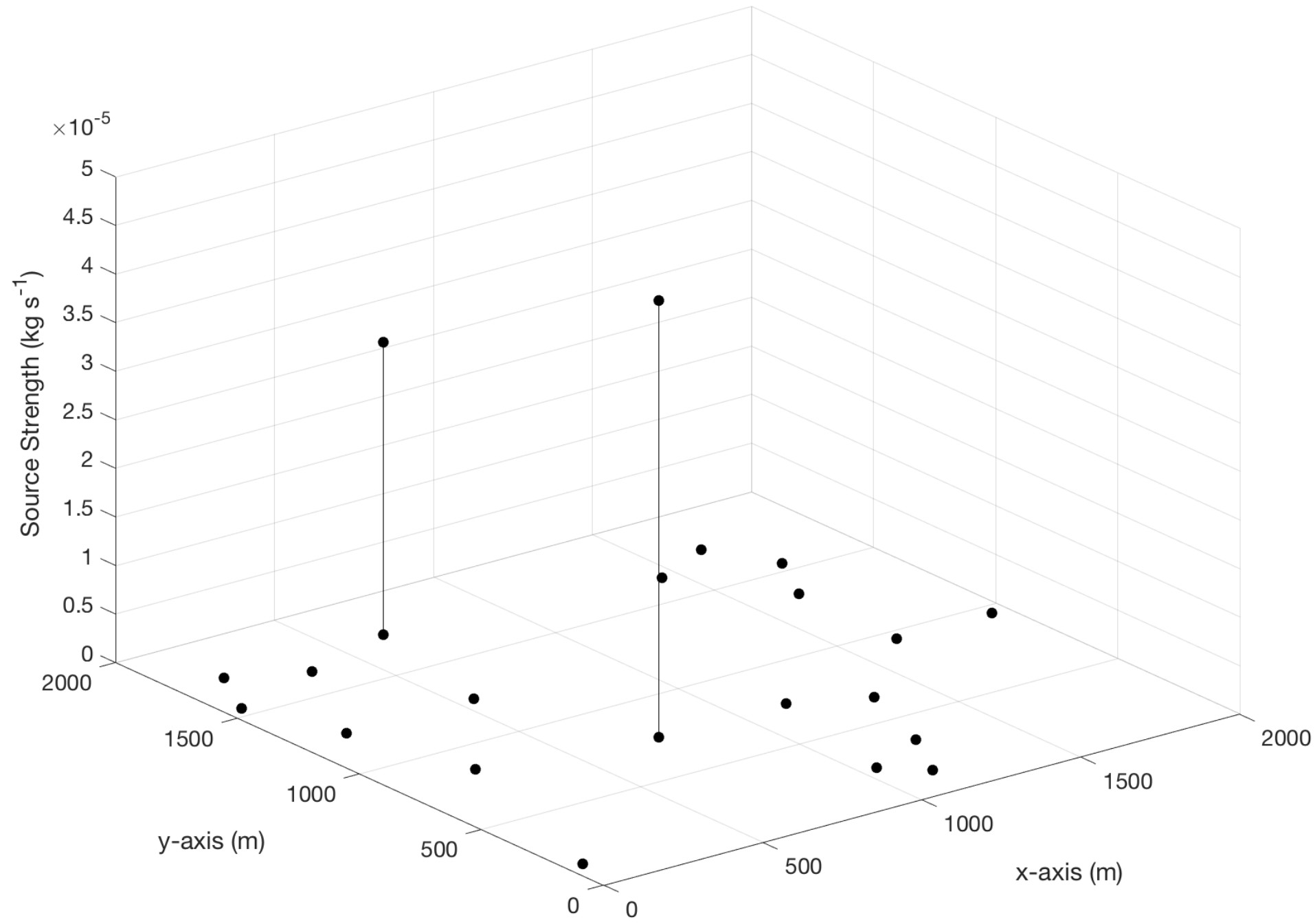

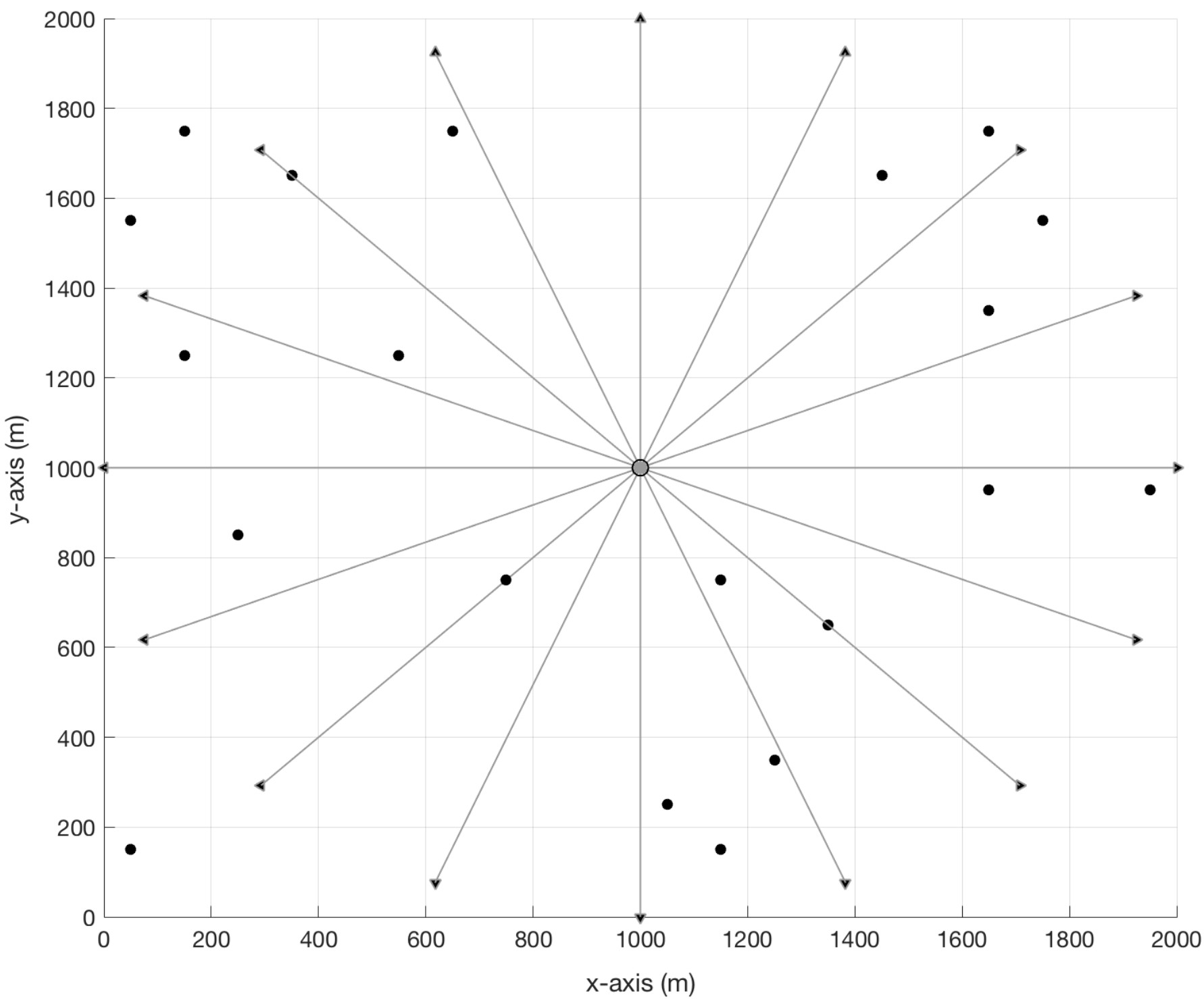

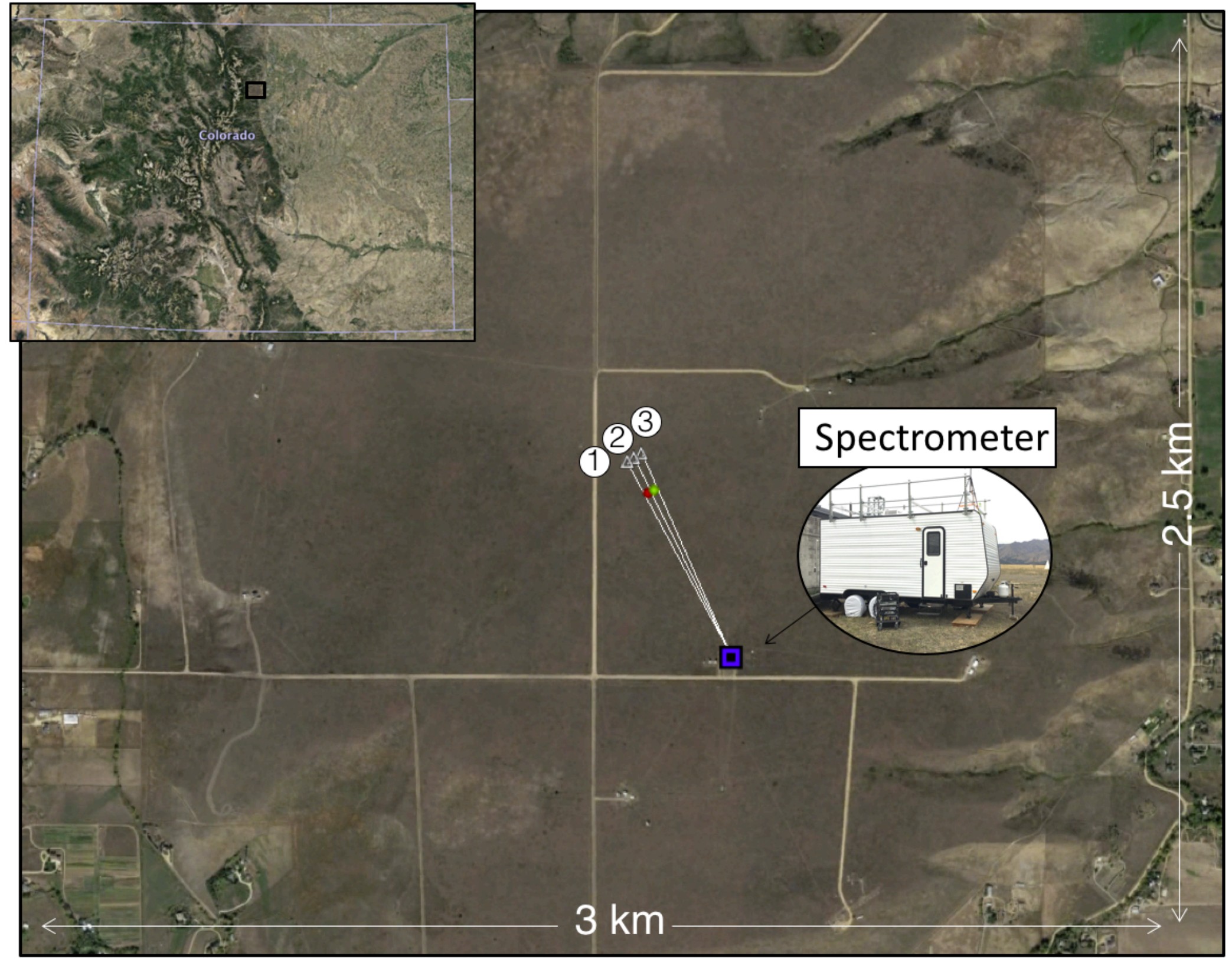

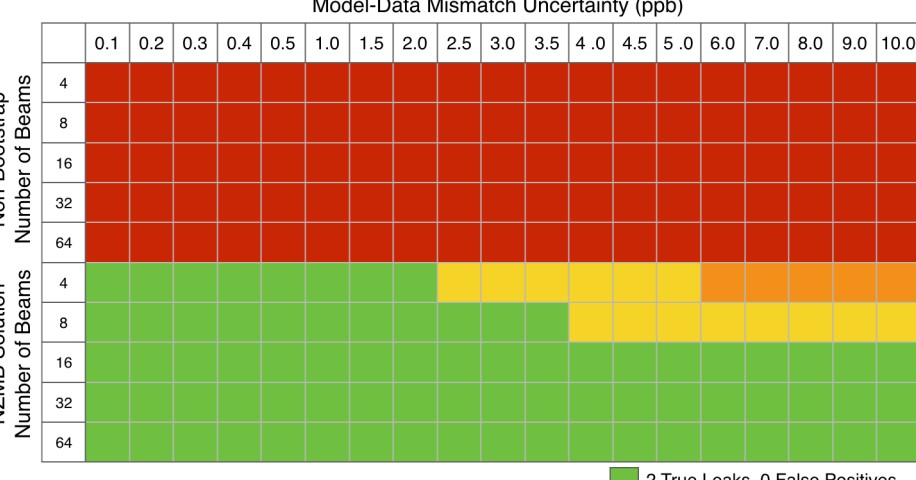

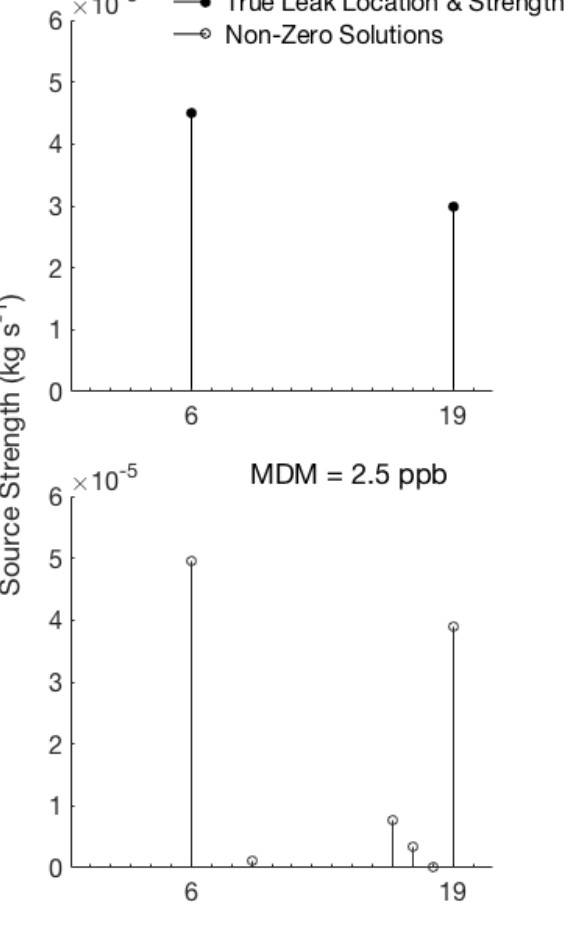
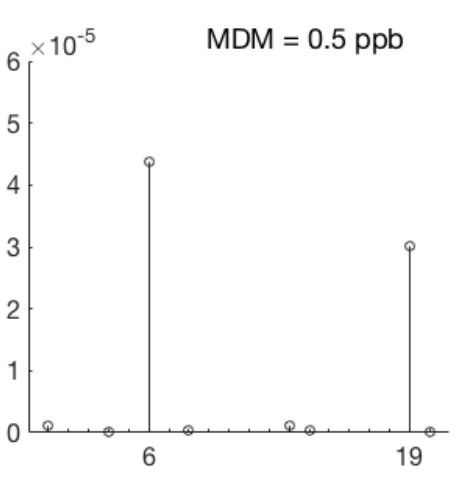
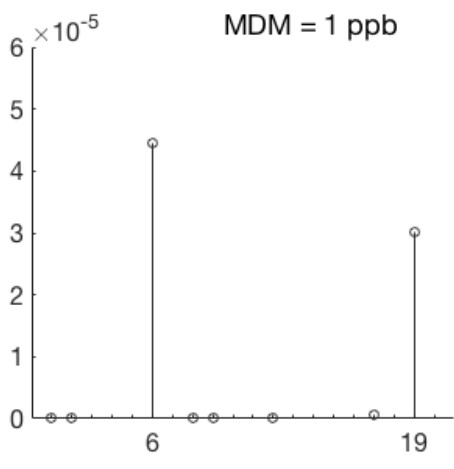
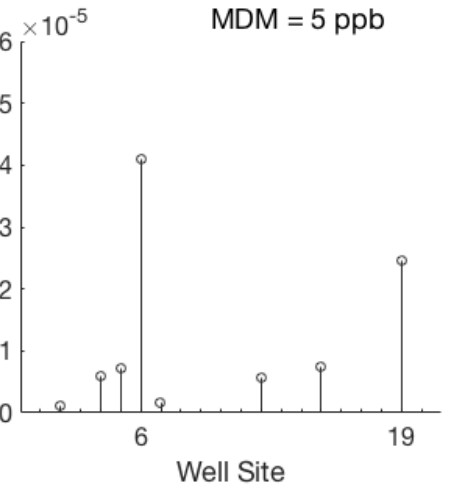
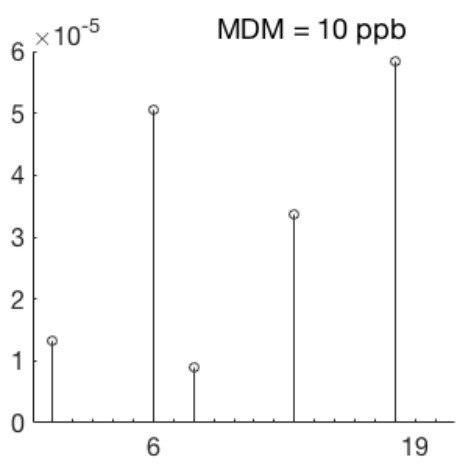

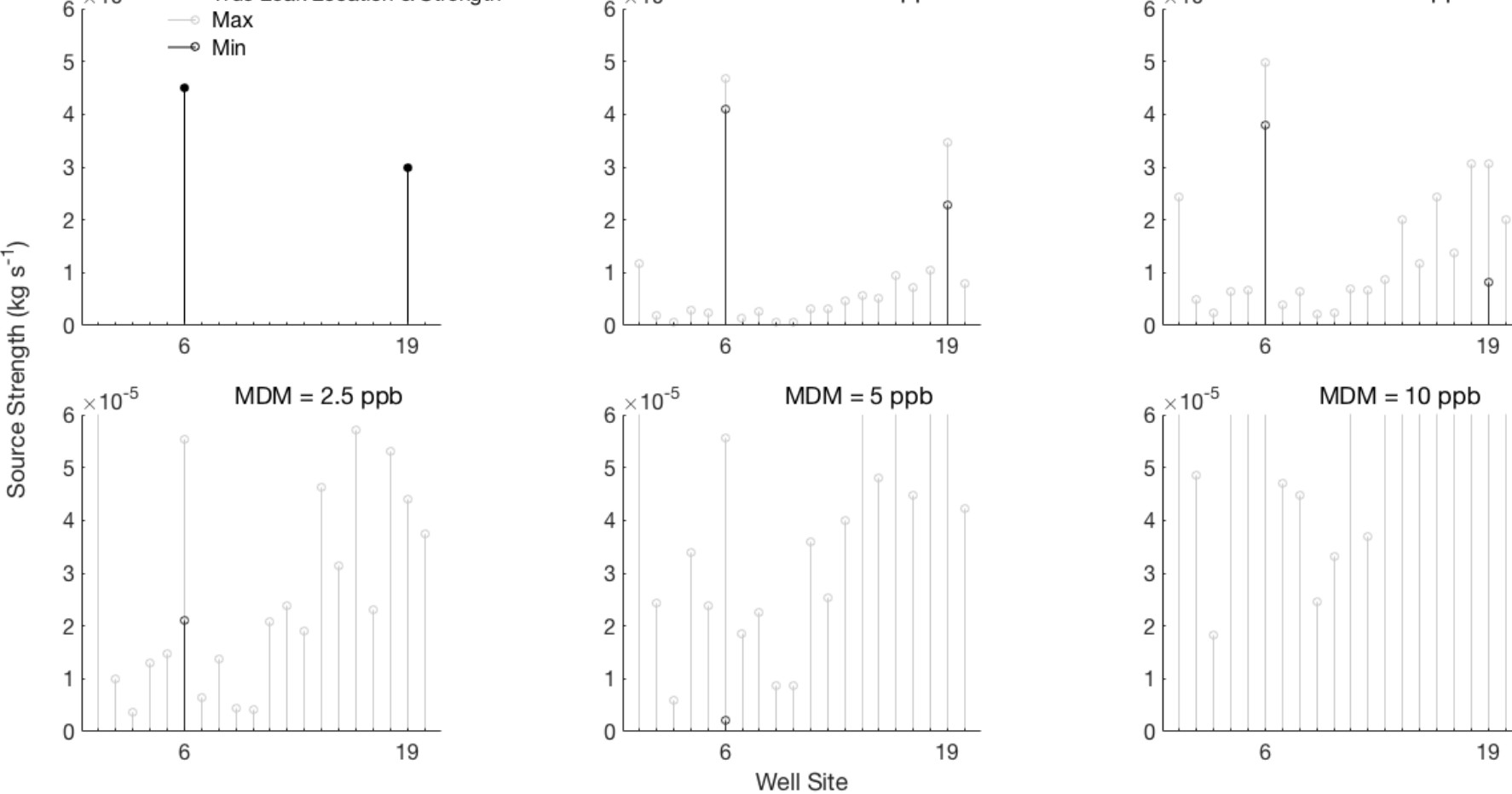

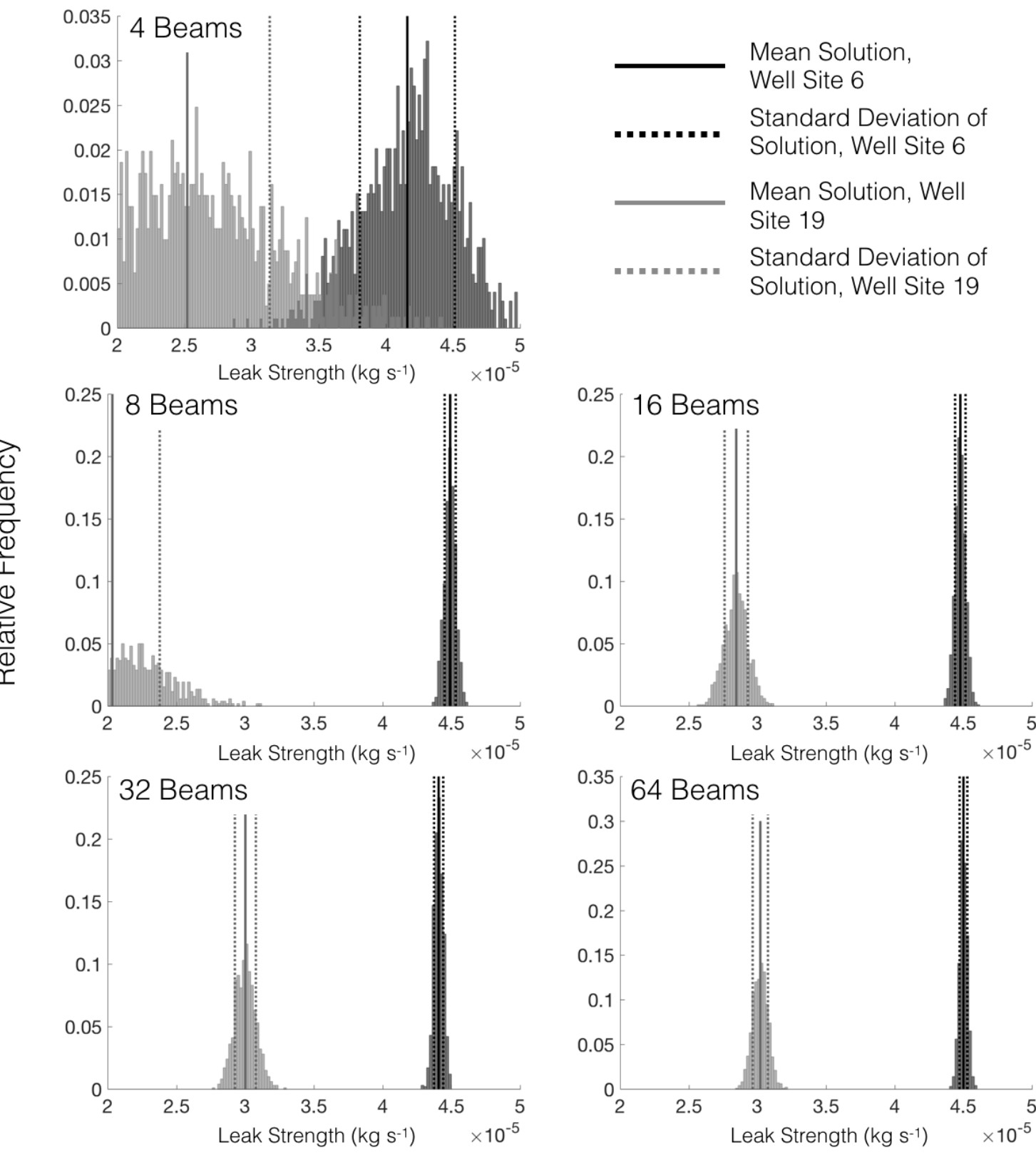

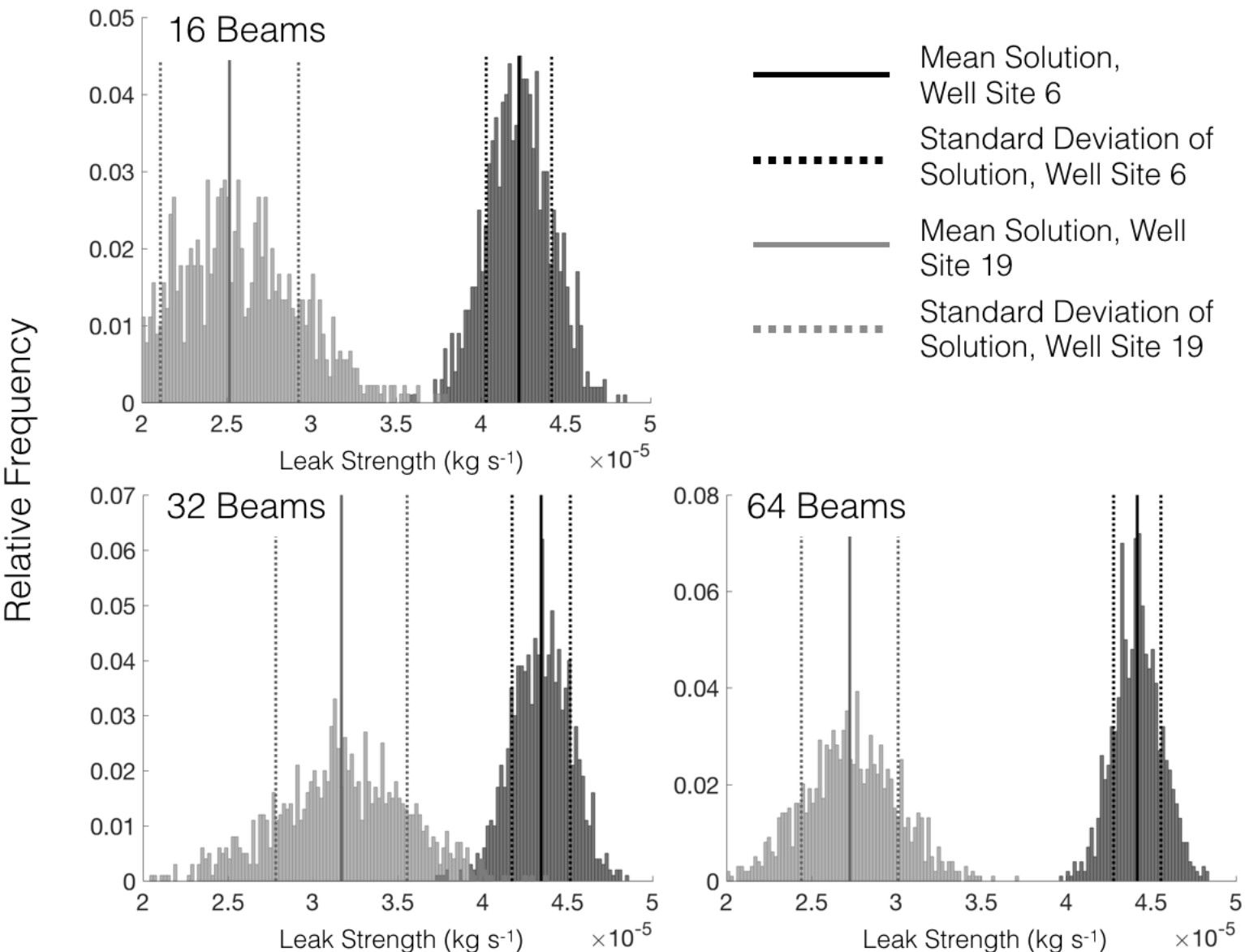

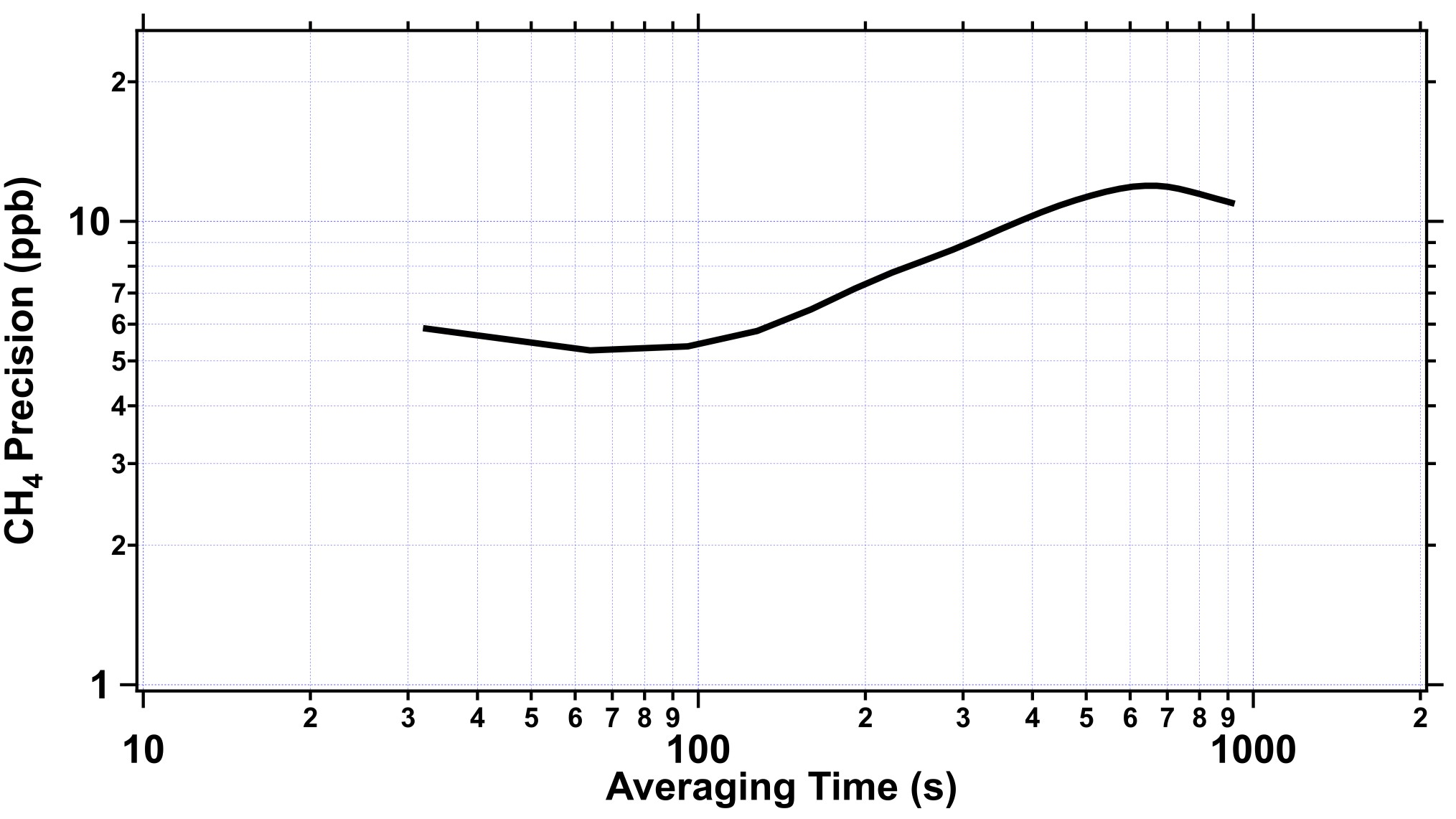

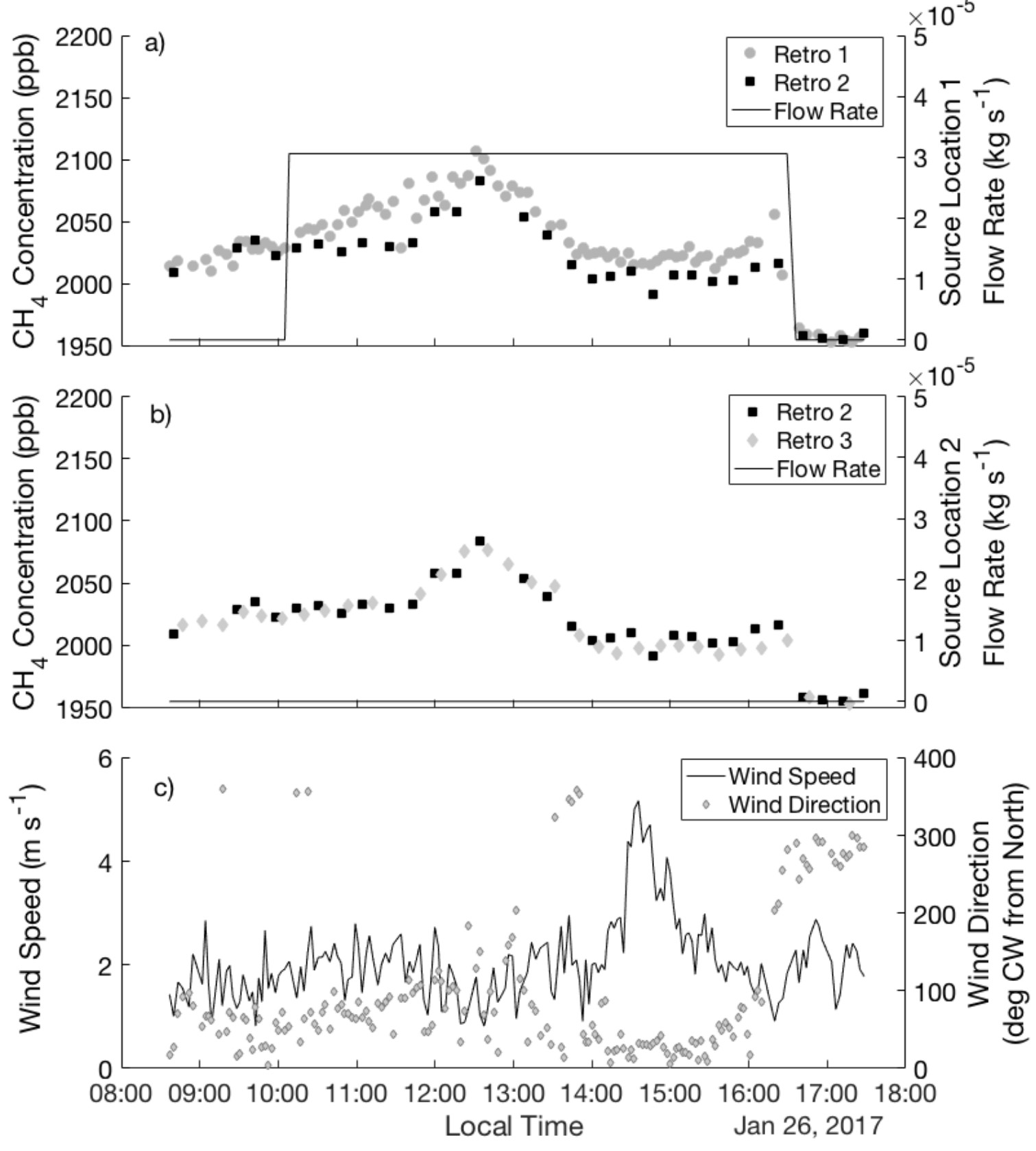

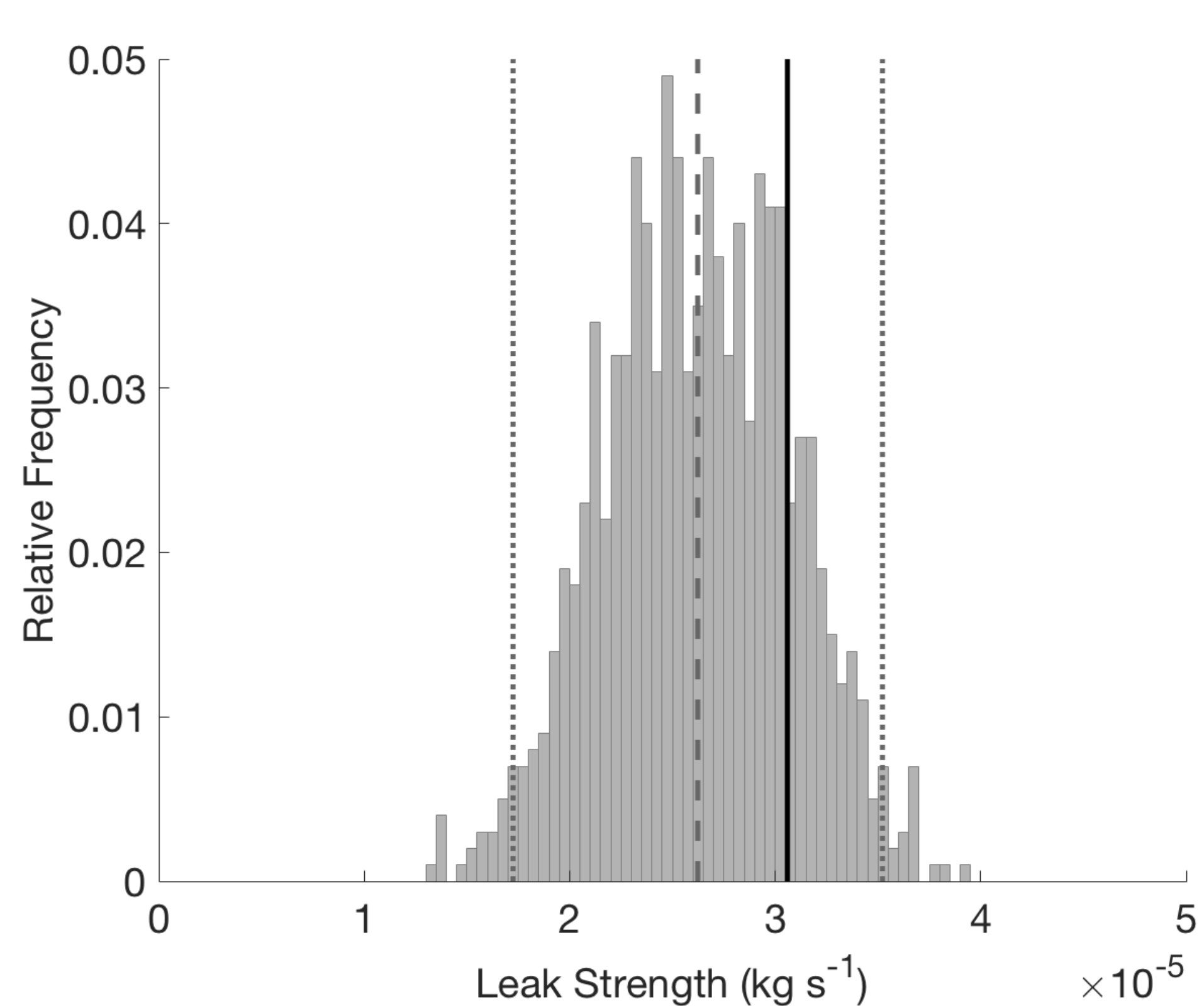

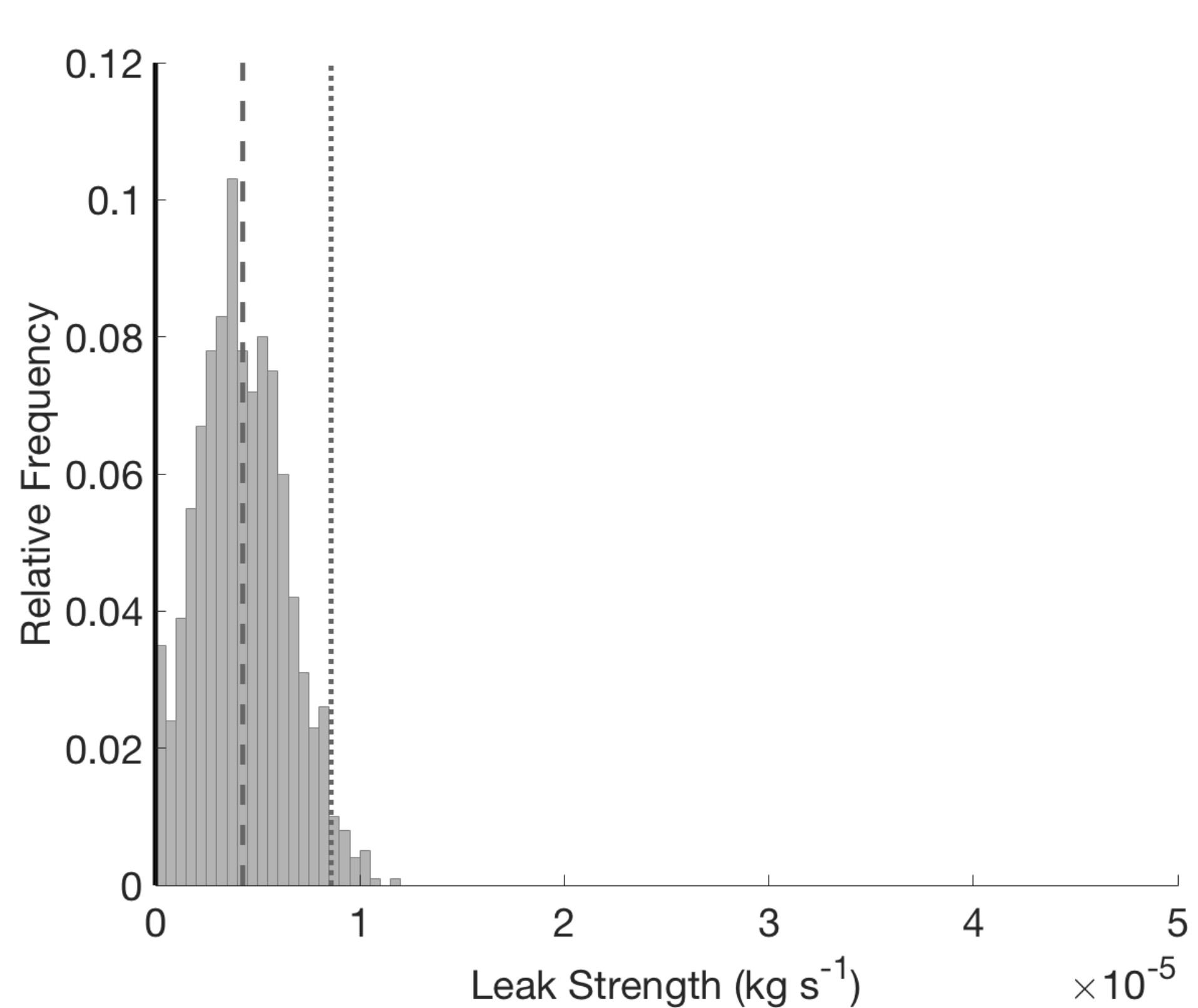