# Peer review of "Methane leak detection and sizing over long distances using dual frequency comb laser spectroscopy and a bootstrap inversion technique"

_Atmospheric Measurement Techniques, 2017_

## Referee Comment (RC1) · Anonymous Referee #2 · 20 Oct 2017

General Comments

In atmospheric "inverse dispersion" problems, we attempt to deduce something about gas transfer to or from the atmosphere by (i) sampling the gas concentration field, (ii) simultaneously gathering sufficient meteorological data to characterize atmospheric transport (mean wind and turbulence), then (iii) invoking an atmospheric dispersion model to make inferences about the source(s). Within that overall scheme there are many variants. Such problems arise and can be solved on every meteorological scale

of atmospheric (or, for that matter, oceanic) motion. Sometimes one knows the exact location of the source(s) and the object is their quantification; sometimes deducing source location (in space and or time) is the most important aim; and sometimes one seeks to deduce source space-time location *and* strength.

This paper is highly relevant to the problem of determining by inverse dispersion the spatial location(s) of point- or near-point sources in the context of gas leak detection (specifically methane) from industrial sites, and an element that is stressed is the avoidance of false positives (for potential leak locations) while not failing to find actual (non-zero) source locations. The authors briefly describe what appears to be a highly capable and useful instrument for their purpose (though describing the instrument is not the key aspect of this paper), but mostly focus on their statistical strategy for optimizing the useful information that can be extracted from their "measurements" (the quotes, because the paper invokes measurements both synthetic and real). I have not been able to fully comprehend that statistical strategy, and I feel the recipe for it can and should be clearer. Some readers may wonder why a Bayesian framework has not been adopted, as perhaps the most rational way to make use of prior information and account for uncertainties.

Setting aside that aspect of the paper, to my mind the most vulnerable aspect of the authors' methodology is their use of the Gaussian plume model as their atmospheric transport paradigm, which treats the turbulent surface layer wind as if it were a regime of unsheared homogeneous turbulence. The authors do recognise that the Gaussian plume model is highly simplistic, and I can accept their argument that its use in the context of their paper is acceptable. But to drive home the importance of the choice of wind model, I must stress that the mean wind speed and the effective eddy difffusivity vary radically with height across the atmospheric surface layer (ASL), in a manner that is well described by Monin-Obukhov similarity theory. There are much higher fidelity models available, and one of these could be substituted without great penalty in terms of computational burden. It is true that at some sites, obstacles or topography may

disturb the transporting wind field such that it is more complex than envisaged even by the better models (some of which are listed below), but they cannot be a worse choice than the Gaussian plume model, which is in effect a mental straitjacket.

Specific Comments

1. The authors' admit that the Gaussian plume model (GPM) is highly simplistic, and I accept their argument that its use in the context of their paper is acceptable. I would disagree with their *categorical* (i.e. no exceptions) assertion that the GPM is "more suitable" if the inversion is based on line-averaged concentration data. The GPM is unnecessarily simple. There are much better analytical solutions that could and should be used to evaluate the $\mathbf{H}$ matrix. Whereas the GPM entirely neglects mean wind shear and treats the eddy diffusivity as constant (the atmospheric surface layer being represented as a regime of unsheared homogeneous turbulence), better models represent the ASL with a mean wind shear and a height-dependent eddy diffusivity that are consistent with the state of the ASL as parameterised by the Obukhov length and friction velocity. These solutions can be rapidly calculated; and where they provide (only) cross-wind integrated concentrations, the authors could easily introduce Gaussian *crosswind* spread (note: it is hard to improve on the assumption of Gaussian crosswind spread without using measured data on wind direction fluctuations).

   A better analytical (or semi-empirical) eddy diffusion model could be sought out from the following references: Philip (1959), Ermak (1977), van Ulden (1978), Nieuwstadt and van Ulden (1978), Huang (1979), Lupini et al. (1981), Wilson (1982), Kormann and Meixner (2001), Sharan & Kumar (2009), Wilson (2015).

2. p5, line 30: Is there an easy argument that the optical detector's response is to line-averaged methane mole fraction as opposed to line averaged mass concentration $[\mathrm{kg\,m^{-3}}]$?

3. At p6 line 5, why "attempts to solve"?

4. I don't understand why the authors contend (p6, line 19) that (in general, with $m$ source locations and $n$ concentrations) their "problem is overdetermined" — do they assume $n > m$?

5. Why (p6 lines 22-23) should it be the case that (or why is it a safe assumption that) "model-data mismatch uncertainty has an un-biased Gaussian distribution"?

6. I find the derivation (p6 line 31 to p7 line 2) hard to follow, yet I suspect it has to be very simple. For instance we have the three symbols $\epsilon_{\mathbf{R}i}$, $\epsilon_{\mathbf{R}}$ and $\epsilon_{bi}$: is the last just the first, in alternative guise?

7. If I have understood correctly, the pool of residual values is a set containing only $n$ members, where $n$ is the number of concentration measurements ($n = 3$ for the field test). Then, for each detector ($i = 1...n$) one randomly draws 1000 samples from that set, with replacement, thereby obtaining a set of 1000 alternative model predictions $y_{bi}$ for (each) source location. The logic for this is not very sound, it seems to me, because all observations are given equal status, irrespectively of their distance from the source(s). In real world cases there could be order-of-magnitude differences in the measured mean concentrations — and indeed in concentration variance and higher moments that, although irrelevant here, surely relate to the trustworthiness or representativeness of a measurement — and in the level of uncertainty in the modelling.

8. At p7, line 13, it might be helpful to be more specific as to what "law of large numbers" means in this context. I expect it amounts to an assumption that the distribution of some mean value (or sum) is Gaussian, even if the numbers being summed do not have a Gaussian distribution (central limit theorem)?

9. At p9 line 21, the authors allude to "model-data mismatch noise." It seems to me that "noise" type errors (which can be dealt with by averaging) are in practice likely

to be far less serious than *systematic* errors arising from the imperfect modelling of atmospheric transport.

10. How is gas-gas interference dealt with? What about detection of "stray" infra-red radiation emitted from the environment and/or from within the telescope?

11. Inversion of the $\mathbf{H}$ matrix can entail severe error if the matrix is "ill-conditioned" as a result of the relative positioning of the sources and detectors: for example if sources are aligned along the wind direction (see Crenna et al. 2008; Flesch et al. 2003).

12. Using the GPM entails selection of appropriate $\sigma$-curves: the choice should be documented.

13. The Conclusion does not, to my mind, sufficiently recognise the potential accuracy gain from using a more sophisticated and realistic atmospheric transport model.

14. I wondered whether the authors are familiar with normal practice in regard to averaging, in micrometeorology. To make sense of surface layer winds of course we necessarily must use statistics, and it is usual to base those on an averaging interval of at least about 15 minutes. However the discussion caused me to suspect that the authors envisage using nearly-instantaneous wind measurements (or, say, 1 min averages) to deduce leak rates from concentration averages over (say) one or two minutes. This may be feasible, indeed it may be a good idea (at the least, it would speed up a search across many potential leak sites). The thing is, however, that one has to recognize that existing, documented, tested, trusted surface layer dispersion models exploit a statistically stable estimate of surface layer state, and that statistical stability demands averaging intervals much longer than a minute (or two). To adapt existing atmospheric transport modes to very

sort averaging intervals one will at the least have to adjust the parameterisation of the eddy diffusivity.

Technical corrections

1. First paragraph of Sec. 2.6.2 uses mixed tenses (present then past).

2. Figure 3 shows only two beams for the field test, whereas the discussion of the experiment alludes to three.

3. The number of panels on Figures (5,6) seems excessive — is it necessary to cover the range in MDM (model-data-mismatch) with such fine steps.

4. The structuring of this paper results in a degree of repetition.

**References**

Crenna, B.P., Flesch, T.K., & Wilson, J.D. 2008. Influence of Source-Sensor Geometry on Multi-Source Emission Rate Estimates. *Atmos. Environ.*, **42**, 7373–7383. DOI: doi:10.1016/j.atmosenv.2008.06.019.

Ermak, Donald L. 1977. An Analytical Model for Air Pollutant Transport and Deposition from a Point Source. *Atmos. Environ.*, **Vol. 11**, 231–237.

Flesch, T.K., Harper, L.A., Desjardins, R.L., Gao, Z., & Crenna, B.P. 2009. Multi-source emission determination using an inverse-dispersion technique. *Boundary-Layer Meteorol.*, **132**, 11–30.

Huang, C.H. 1979. A Theory of Dispersion in Turbulent Shear Flow. *Atmos. Environ.*, **13**, 453–463.

Kormann, R., & Meixner, F.X. 2001. An Analytical Footprint Model for Non-Neutral Stratification. *Boundary-Layer Meteorol.*, **99**, 207–224.

Lupini, Renzo, & Tirabassi, Tiziano. 1981. A Simple Analytic Approximation of the Ground-Level Concentration for Elevated Line Sources. *J Appl Meteorol*, **20**, 565–570.

Nieuwstadt, F. T. M., & Ulden, A. P. Van. 1978. A Numerical Study on the Vertical Dispersion of Passive Contaminants from a Continuous Source in the Atmospheric Surface Layer. *atmos. envt*, **12**, 2119–2124.

Philip, J. R. 1959. The Theory of Local Advection. *J. Meteorol.*, **16**, 535–547. (I have clarifying correspondence with JRP in a folder.).

Sharan, M., & Kumar, P. 2009. An Analytical Model for Crosswind Integrated Concentrations Released from a Continuous Source in a Finite Atmospheric Boundary Layer. *Atmos. Environ.*, **43**, 2268–2277.

van Ulden, A. P. 1978. Simple Estimates for Vertical Diffusion From Sources Near the Ground. *Atmos. Environ.*, **Vol. 12**, 2125–2129.

Wilson, J.D. 1982. An Approximate Analytical Solution to the Diffusion Equation for Short-Range Dispersion from a Continuous Ground-Level Source. *Boundary-Layer Meteorol*, **23**, 85–103.

Wilson, J.D. 2015. Dispersion from an area source in the unstable surface layer: an approximate analytical solution. *Q.J.R. Meteorol. Soc.*, **141**(693), 3285–3296.

---

## Referee Comment (RC2) · Anonymous Referee #1 · 13 Nov 2017

General comments

The paper proposes a special observing system and statistical analysis to continuously detect and monitor methane leaks from gas production sites. The suggested method combines line-averaging concentration measurements with an atmospheric transport model and a novel statistical method to derive the spatially dependent methane concentration. In this way, the paper presents innovative concepts and fits within the scope of AMT.

[Figure]

The introduction gives a clear and concise motivation for the study, however related line-averaging measurement techniques and their advantages and disadvantages are not sufficiently addressed (e.g. Open-Path Tunable Diode Laser OP-TDL and Open-Path Fourier transform infrared spectroscopy OP-FTIR). Additionally, the non-zero minimum bootstrap (NZMB) method is introduced without explaining the motivation behind this development in comparison to other possible methods of inverse data analysis. An extended description of the state of the art is necessary to evaluate and classify the new measurement and analysis method.

The atmospheric transport model is one of the central methods of the data analysis. The authors apply a Gaussian plume model assuming a constant methane source through the time for the synthetic data tests and for the field data under similar conditions. This approach and the application of a Gaussian plume model for the purpose of continuous methane leak detection in a real environment are extremely questionable. The Gaussian plume model for atmospheric dispersion is assuming steady-state air concentrations and mass fluxes. There are many references that show the applicability of the Gaussian model: it is primarily used to calculate seasonal or annual statistical values of air concentrations near the ground. It is recommended to use more sophisticated and realistic transport models to ensure a general applicability of the proposed method in the future. The measurement and analysis concept has a high potential of applicability if weak points of single methods will be eliminated, especially regarding the atmospheric transport model.

Specific comments

1. Introduction: It is highly recommended to give proper credit to related work regarding the measurement technique (line-averaging measurement methods to derive gas concentrations, e.g. OP-FTIR) and the statistical analysis methods.

2. p. 2, l. 8: The 'high global warming potential' should be quantified (in comparison to carbon dioxide) to further explain the strong motivation for this study.

[Figure]

3. p. 2, l. 9: What is the reference value for the threshold leak rate in percent (3.2%)?

4. p. 2, l. 18: 'Cold temperatures' is a rather colloquial expression, better: lower temperature values.

5. p. 3, l. 1: What does it mean 'agreement of measurements under different conditions'? Which range of air temperature/pressure values? Which range of atmospheric stability and turbulence conditions?

6. p. 3, l. 2: What does it mean 'long periods of time'? How long is it?

7. p. 3, l. 4: What is the 'measurement uncertainty'? Give a short explanation of this quantity: Is it a statistical value (estimated from the standard deviation) or the technical uncertainty (depending on the device) or the analysis uncertainty or…?

8. p. 3, l. 4: What does it mean 'long pathlengths'? Please be more precise or give an example or a range.

9. p. 3, l. 4: Are the data and results of Coburn et al. freely available and reproducible in the meantime?

10. p. 4, l. 5: Is the applicability of the method influenced by the special environmental conditions (e.g., wind speed and direction, atmospheric stratification) of this day in January, 2017?

11. p. 4, Gaussian plume atmospheric transport model: The application of a Gaussian plume model for the purpose of continuous methane leak detection in a real environment is extremely questionable because the model assumes steady-state air concentrations and mass fluxes. The Gaussian model maybe applied under the limited conditions of the synthetic and the real-world experiments described by the authors. However, the general applicability of the proposed measurement and analysis method is limited due to the atmospheric transport model. The cited paper of Leuning et al. (2008) contains one example of a more realistic dispersion model, the Lagrangian stochastic (LS) dispersion analysis. LS analysis can be used when the assumptions

of constant and homogeneous turbulence (e.g., eddy diffusivity) are not satisfied. In contrast to the Gaussian plume model, LS models incorporate wind shear. It is highly recommended to use such a kind of transport model to allow a general and continuous applicability of the proposed measurement and analysis method under different environmental conditions.

12. p. 5, l. 22: What does the term '$(c/x)modelled$' describes, the relationship between the point source emission and the line-averaged concentration (=spectrometer beam)? Please explain the relationship between the different scales: point source (emission) – concentration and line-averaged measurements of concentration together with the atmospheric transport model.

13. p. 5, section 2.2: Which spatial and temporal resolution for the derived gas concentration can be achieved using this method? How much is the environmental influence (e.g., background air temperature)?

14. p. 6, l. 4: Which fluxes? Please describe values more specific.

15. p. 6, l. 7: Is the system rather overdetermined or underdetermined for your examples? See also l. 19. The system is overdetermined including uncertainties/errors into the system, really?

16. p. 6, l. 17 and p. 7, l. 16: How did you estimate and quantify the several kinds of uncertainty? Did you only use the standard deviation as a value for the statistical uncertainty? It is highly recommendable to calculate the total uncertainty of the combined measurement and analysis method (uncertainty due to devices, measurement methods, transport model, inversion model. . .).

17. p. 8, l. 4: What does it mean 'idealized scenario', homogeneous wind fields (wind speed and direction)?

18. p. 8, l. 7: How long is the sampling time of each beam?

19. p. 9, l. 2: Which influence of atmospheric stability can be expected?

20. p. 9, section 2.5.4: An enhanced estimation of all parts of total uncertainty is necessary to evaluate the applied range of model-data mismatch values.

21. p. 10, l. 9: Please give an example for 'long periods of time'.

22. p. 11, l. 1: It can be expected that a local wind circulation is developing at Table Mountain. The assumption of homogeneity and stationarity of turbulent fields is highly questionable under these conditions (Gaussian plume model for atmospheric transport). It has to be checked that the assumptions of the Gaussian plume model are valid for the investigated location and time period.

23. p. 11, l. 5: Can you give a reference for the applied threshold of 0.8 m/s?

24. p. 14, l. 2: The (short) averaging time of 1-2 min disagrees probably with the assumption of stationarity of the Gaussian plume model. It is questionable if the assumptions of atmospheric transport model are satisfied. At least, an enhanced contribution of uncertainty should be included in the analysis (and conclusions).

25. p. 16, section 6: The critical discussion of the used atmospheric transport model is missing here. If a simple assumption (Gaussian plume model) is used for real (complex) environmental conditions, a comprehensive analysis of uncertainties is needed to evaluate the potential and the general applicability of the method.

26. Fig. 10: Are the concentration data referred to the length of the beam? Please provide the wind direction according to meteorological conventions: 0 deg = north, 180 deg = south, 90 deg = east, 270 deg = west). Which averaging time was used to provide the data?

Technical corrections

1. p. 3, l. 4: 'Coburn' – 'et al.' is missing.

2. p. 5, l. 28: 'Truong et al.; Waxman et al.' – The year is missing.

3. p. 5, l. 32: 'Coburn' – 'et al.' is missing.

4. p. 5, l. 33: 'tempus lorem' – misprint?

5. p. 8, l. 8: 'that that' – delete one 'that' 6. Table 2: Source Location 1 and NZMB solution: unit must be kg/s

---

## Author Comment (AC1) · 9 Dec 2017

December 9, 2017

Dear Reviewer and AMT Editorial Office,

Thank you for your detailed reading of our manuscript and the constructive suggestions you have offered for its improvement. Your review and analysis have been exceedingly helpful. We hope that we have understood and incorporated your critiques, and, in accordance with them, improved the analysis and the presentation of our work.

Please see below a detailed response to each of your suggestions and a description of how we improved our study to address your concerns. Reviewer comments are in black and our responses are in blue. Line numbers in our responses below refer to the mark-up draft of the new manuscript.

In addition to the changes in response to the reviewer's comments and suggestions, several additional changes have been made to the manuscript. These include: 1) a small correction to the flow rates for the field test data that has been discovered to be necessary since the initial submission; 2) a slight change to the method in which background methane is calculated that leads to fewer sample losses due to data gaps; and 3) resolution of a recently identified issue with beam length scaling, which has led to very small adjustments to the raw concentration data. None of these changes have altered the outcome or scientific message of the paper.

The authors are very grateful for your time and energy.

Sincerely,

Dr. Caroline Alden and co-authors
* * *
Below are detailed responses to the suggestions and comments made by the reviewers. The reviewer comments are in black and our responses are in blue.

Anonymous Referee #1:

General comments
The paper proposes a special observing system and statistical analysis to continuously detect and monitor methane leaks from gas production sites. The suggested method combines line-averaging concentration measurements with an atmospheric transport model and a novel statistical method to derive the spatially dependent methane concentration. In this way, the paper presents innovative concepts and fits within the scope of AMT.

The introduction gives a clear and concise motivation for the study, however related line-averaging measurement techniques and their advantages and disadvantages are not sufficiently addressed (e.g. Open-Path Tunable Diode Laser OP-TDL and Open- Path Fourier transform infrared spectroscopy OP-FTIR). Additionally, the non-zero minimum bootstrap (NZMB)

method is introduced without explaining the motivation behind this development in comparison to other possible methods of inverse data analysis. An extended description of the state of the art is necessary to evaluate and classify the new measurement and analysis method.

The atmospheric transport model is one of the central methods of the data analysis. The authors apply a Gaussian plume model assuming a constant methane source through the time for the synthetic data tests and for the field data under similar conditions. This approach and the application of a Gaussian plume model for the purpose of continuous methane leak detection in a real environment are extremely questionable. The Gaussian plume model for atmospheric dispersion is assuming steady-state air concentrations and mass fluxes. There are many references that show the applicability of the Gaussian model: it is primarily used to calculate seasonal or annual statistical values of air concentrations near the ground. It is recommended to use more sophisticated and realistic transport models to ensure a general applicability of the proposed method in the future. The measurement and analysis concept has a high potential of applicability if weak points of single methods will be eliminated, especially regarding the atmospheric transport model.

We thank Referee 1 for the time that they have taken in carefully reviewing this work and for their comments. In our response to the reviewer's specific comments below, we address the points raised in the paragraph above: the need for discussion of other open-path measurement techniques, and the need for discussion of the appropriateness of a steady-state transport model (namely the Gaussian plume model) for this work.

Additionally, the reviewer emphasizes a need for additional discussion of how the NZMB method fits into the context of existing inverse data analysis techniques. We have added new discussion of the state of existing methods for source location to the Introduction Section (page 3, line 28-33). Bootstrapping techniques and least-squares fitting techniques have been used for many applications in the sciences. However, there is not, to our knowledge, a field of inverse analysis devoted to minimization of false-positive sources. In response to the reviewer's suggestion, we have added introductory motivation for the development of the NZMB method to the beginning of Section 2.4 (page 7, lines 11-13). In the text we emphasize references to the traditional methods (least-squares fitting techniques and bootstrap analysis) that are combined here to produce a novel statistical methodology.

Specific comments

1. Introduction: It is highly recommended to give proper credit to related work regarding the measurement technique (line-averaging measurement methods to derive gas concentrations, e.g. OP-FTIR) and the statistical analysis methods.

Mentions and citations of other open-path laser techniques have been added to the Introduction in paragraph 3 (page 3, lines 3-5). As the focus of this paper is not the measurement technique itself, but rather the statistical methods to use line-integrated measurements as a tool for source attribution, we have kept this added discussion relatively brief while taking care to reference important previous work that does focus on the measurement technique (and which covers previous line-averaging methods in greater depth).

See response to the above comment for discussion of reference to previous work regarding the statistical analysis methods.

2. p. 2, l. 8: The 'high global warming potential' should be quantified (in comparison to carbon dioxide) to further explain the strong motivation for this study.

The GWP of $CH_4$ has been added to the sentence referenced by the reviewer in Section 1 (page 2, lines 8-10).

3. p. 2, l. 9: What is the reference value for the threshold leak rate in percent (3.2%)?

The reference for the threshold leak rate cited is Alvarez et al. (2012). We have added clarification and a second reference to that publication in the sentence referenced by the reviewer (page 2, line 10), in order to make it clear which of the two references at the end of that sentences pertains to the cited rate of 3.2%.

4. p. 2, l. 18: 'Cold temperatures' is a rather colloquial expression, better: lower temperature values.

The sentence cited by the reviewer (Section 1, page 2, line 20) has been edited to reflect this improvement.

5. p. 3, l. 1: What does it mean 'agreement of measurements under different conditions'? Which range of air temperature/pressure values? Which range of atmospheric stability and turbulence conditions?

The paragraph referred to by the reviewer (Section 1, paragraph 3) has been edited for clarity in this regard (see page 3, lines 5-14). In particular, we have included reference to a 2017 study by Waxman et al., who found that measurements made by two independent dual frequency comb spectrometers measuring the same air masses over a period of 2 weeks agreed under a range of ambient temperatures (4.6 – 28.9 ºC outdoors and 17 – 25 ºC indoors; the instruments were in a room that was open to the outside), and a range in relative humidity of 10-90%. A range of pressure, stability and turbulence conditions were also experienced. Furthermore, we have edited the text to clarify the principle reason why high measurement reproducibility should extend beyond ambient conditions to more extreme conditions, given that the retrieval is dependent on the quality of the absorption models and not on ambient conditions.

6. p. 3, l. 2: What does it mean 'long periods of time'? How long is it?

While the instrument does not require calibration and measurements can be directly cross-compared over periods of weeks to months (and likely years), placing an exact quantity on this length of time is involved and would require extensive additional study and experimentation. We therefore chose to simplify this statement instead. We have removed the phrase "long periods of time" and replaced it with "different conditions and times" (page 3, lines 8-9). To offer additional clarity on this point, we have also added the sentence: "Previous work also demonstrates that this method of atmospheric trace gas measurement does not require regular or

traditional calibration (Coburn et al., n.d.; Rieker et al., 2014; Truong et al., 2016; Waxman et al., 2017)" (page 3, lines 14-16).

7. p. 3, l. 4: What is the 'measurement uncertainty'? Give a short explanation of this quantity: Is it a statistical value (estimated from the standard deviation) or the technical uncertainty (depending on the device) or the analysis uncertainty or. . .?

The reviewer makes an important point that, as written, it was not clear which metric for uncertainty was being described. We have edited the text in question to clarify that the spectrometer demonstrates high measurement precision (page 3, line 17).

8. p. 3, l. 4: What does it mean 'long pathlengths'? Please be more precise or give an example or a range.

The text has been updated to reflect the length of the spectrometer paths (1 km, one-way) (page 3, lines 17-18).

9. p. 3, l. 4: Are the data and results of Coburn et al. freely available and reproducible in the meantime?

The Coburn et al. results and data are freely available and reproducible. The Coburn et al. manuscript is posted on the Cornell University Library's open access e-prints service, arXiv.org. The manuscript can be found at the following link: https://arxiv.org/abs/1711.08067.

10. p. 4, l. 5: Is the applicability of the method influenced by the special environmental conditions (e.g., wind speed and direction, atmospheric stratification) of this day in January, 2017?

The choice of January 26, 2017 was independent of wind conditions and instead depended on the preparedness of the team and instrumentation for field deployment. We have edited the text to emphasize that there were no special environmental conditions on this day (page 4, lines 22-24). Examination of long-term (1 year) mean meteorological conditions at a nearby weather station (KCOLONGM30) demonstrates that the wind speed and wind direction on this day were close to average.

11. p. 4, Gaussian plume atmospheric transport model: The application of a Gaussian plume model for the purpose of continuous methane leak detection in a real environment is extremely questionable because the model assumes steady-state air concentrations and mass fluxes. The Gaussian model maybe applied under the limited conditions of the synthetic and the real-world experiments described by the authors. However, the general applicability of the proposed measurement and analysis method is limited due to the atmospheric transport model. The cited paper of Leuning et al. (2008) contains one example of a more realistic dispersion model, the Lagrangian stochastic (LS) dispersion analysis. LS analysis can be used when the assumptions of constant and homogeneous turbulence (e.g., eddy diffusivity) are not satisfied. In contrast to the Gaussian plume model, LS models incorporate wind shear. It is highly recommended to use such

a kind of transport model to allow a general and continuous applicability of the proposed measurement and analysis method under different environmental conditions.

We agree with the reviewer that the simplicity of the Gaussian plume model with respect to assumptions of constant and homogeneous turbulence may not be as useful for field tests as, for example, a Lagrangian stochastic dispersion approach. We have added discussion of this point to Section 2.1 (page 5, lines 17-22). We have also added a discussion of this subject to the conclusions section of the paper, Section 6 (page 20, lines 20-31), which addresses these caveats to using the Gaussian plume model for representations of atmospheric dispersion.

With respect to the use of a steady-state model such as the Gaussian plume model for the present application, we are grateful for the feedback on the clarity of this point. For cases in which the time of transport from the source to the "receptor" (measurement location) is comparable to the time over which the data is averaged, a steady-state model such as the Gaussian plume model is appropriate (Gifford, 1976). In our case, the measurement averaging time is 120 seconds. In the field test cases, the mean distance between all beam locations (discretized into 100 equal-length sections) and the leaks they are monitoring, divided by the mean wind speed on 1/26/2017 of 2.1 m s$^{-1}$, equals a mean travel time of 100 seconds. The travel time from beam to receptor is therefore comparable to the averaging time for measurements, satisfying the requirements necessary for the Gaussian plume model assumptions of steady-state to be valid. Note that for the synthetic data cases there is no time dimension, so the use of a steady-state model is not an issue. To help better convey this point, we have added the parenthetical clarification that the plume model characterizes the steady-state in the first paragraph of Section 2.1 (page 5, line 14), and have added further clarification on page 5, lines 26-31.

For the experiments here, the assumption of constant mass fluxes is also valid, as the controlled releases of methane were constant. Future studies of intermittent leaks will need to assess transport as a function of the timescale of intermittency.

12. p. 5, l. 22: What does the term '(c/x)modelled' describes, the relationship between the point source emission and the line-averaged concentration (=spectrometer beam)? Please explain the relationship between the different scales: point source (emission) – concentration and line-averaged measurements of concentration together with the atmospheric transport model.

Section 2.1 (page 6, lines 8-15) has been edited for clarity on this point. Further, we have added a parenthetical note referring readers to Section 2.5.3 for a detailed explanation of the scaling between point source emission, to point source concentration, to line-averaged concentration (page 6, lines 14-15).

13. p. 5, section 2.2: Which spatial and temporal resolution for the derived gas concentration can be achieved using this method? How much is the environmental influence (e.g., background air temperature)?

The temporal resolution for the derived gas concentration depends on averaging time, as is shown by the Allan deviation in Figure 9 and described in Section 4.1. As averaging time increases, measurement precision increases, until such time that atmospheric variability begins to

erode measurement repeatability. We have added a sentence describing this characteristic of the DCS measurement to Section 2.2, along with a reference to Section 4.1, where it is discussed in more detail (page 6, lines 23-25).

The spatial resolution scales with beam length (which is easily adjustable by moving retroreflectors closer to or further away from the spectrometer), and beam width (which scales with telescope diameter). A discussion of this concept has been added to Section 2.2 (page 6, lines 25-27).

A discussion of the influence of environmental conditions has been added to Section 1 (page 3, lines 9-14). The addition of this text is described in greater detail in the author response to "Specific Comment 5", above.

14. p. 6, l. 4: Which fluxes? Please describe values more specific.

The paragraph referenced in Section 2.3 (page 6, line 29 – page 7, line 2) has been edited to reflect the requested improvements to specificity. Additionally, clarification has been added to Section 2.1 (page 6, lines 8-15), which is where the vector mentioned by the reviewer is described (that referenced in Section 2.3). A reference to that description has also been added to Section 2.3 (page 6, lines 31-32).

15. p. 6, l. 7: Is the system rather overdetermined or underdetermined for your examples? See also l. 19. The system is overdetermined including uncertainties/errors into the system, really?

The sentence referenced in section 2.4 has been edited to clarify the definition of overdetermined in this context: that is, that $n$ is greater than $m$ (page 7, line 22). Multiple measurements along each beam are used in each fit, so that $m$ is never less than $n$.

The following, additional changes to the manuscript have been made to further clarify the dimensions of $n$ and $m$:

In Section 2.1 (page 6, lines 10-11), the sentence was added: "In the synthetic tests and field tests described here, multiple measurements are made along each beam, such that $n$ is always greater than $m$."

In Section 2.5.1, the value of $m$ for each test is now explicitly defined ($m = 20$) (page 9, lines 17-18).

In Section 2.5.3 (page 10, lines 29-31 ), the following sentences were added: "In the synthetic tests, the dimensions of $n$ (e.g., the length of the atmospheric concentration vector, $\mathbf{c}$) vary along with the number of beams per spectrometer-detector system and the number of meteorological conditions. In the configuration of 4 beams, for example, $n = 216 * 4$, because each distinct meteorological condition is applied to each beam. In the 8 beam configuration $n = 216 * 8$, in the 16 beam configuration $n = 216 * 16$, and so on."

A reference to the above section has also been added to Section 2.5.4 (page 11, line 21).

In Section 2.6.2 (page 12, lines 21-23), the dimensions of *m* are now specified for each case (in both cases, $m = 1$).

In Section 2.6.5 (page 13, lines 23-30), a new paragraph has been added that explicitly names the values of *n* for both field test cases: that for source location 1 and that for source location 2. For source location 1, all downwind measurements along beams (retros) 1 and 2 are used in the least squares fit, such that $n = 63$. For source location 2, all downwind measurements along beams (retros) 2 and 3 are used, such that $n = 30$.

16. p. 6, l. 17 and p. 7, l. 16: How did you estimate and quantify the several kinds of uncertainty? Did you only use the standard deviation as a value for the statistical uncertainty? It is highly recommendable to calculate the total uncertainty of the combined measurement and analysis method (uncertainty due to devices, measurement methods, transport model, inversion model. . .).

The reviewer brings up an important need for clarification on the subject uncertainty.

On page 6, line 17 (of the original manuscript), the reviewer points to the role of model-data mismatch uncertainty in the NZMB method. We agree that the original manuscript did not adequately explain that this source of uncertainty is *not* estimated by the user in the traditional "bottom-up" sense. That is, in many inversion frameworks, it is necessary to estimate (or in the reviewer's words, "calculate") "the total uncertainty of the combined measurement and analysis method". In the NZMB method, there is no role for "bottom-up" uncertainty assessments. Instead, the empirical fit to the data is used as the uncertainty estimate. This method inherently encompasses all of the sources cited by the reviewer (measurement device and technique, transport, representation errors, etc.) but uses the fit to the data as an estimate of such rather than a "bottom-up" estimation relying on a priori knowledge of instrumental, transport and modeling constraints. In order to address the helpful suggestion by the reviewer for clarity on this point, we have added clarifying language to Section 2.4 (page 7, lines 24-27) of the revised manuscript.

On Page 7, line 16 (and we assume also lines 17-18) (of the original manuscript) the reviewer points to the method for assessing flux uncertainty in the NZMB method. We agree that here, also, a better justification for this method of uncertainty quantification is needed. Here again, we use the empirically derived distribution of the fit to the data to assess uncertainty in the flux strength, which, as the reviewer points out, is largely a function of instrumental measurement uncertainties, transport model uncertainties, and model uncertainties. The law of large numbers justifies that when the number of bootstrap operations is large, the distribution of the bootstrapped leak strength approaches the probability distribution of the leak strength. We have added text that clarifies this point and outlines the statistical justification for this method of assessing flux uncertainty to Section 2.4 (page 8, line 33 – page 9, line 3).

17. p. 8, l. 4: What does it mean 'idealized scenario', homogeneous wind fields (wind speed and direction)?

This sentence has been revised for clarity (Section 2.5.2, page 9, lines 24-26). The text now explains why a sampling of different wind conditions on each beam is an ideal scenario for generating a large population of independent measurements.

18. p. 8, l. 7: How long is the sampling time of each beam?

In the synthetic data tests, there is no time dimension. In the field-data tests, each beam is sampled for 2 minutes.

19. p. 9, l. 2: Which influence of atmospheric stability can be expected?

The influence of parameterizations of atmospheric stability on the results of the synthetic data tests were not investigated. This is because in these tests, following conventions of most observing system simulation experiments, transport is treated as a perfectly-known characteristic of the experiment. That is, changes in the stability parameter used in the calculations for the synthetic data tests would not be expected to have an impact on the results of the study, because in these experiments the same transport characteristics that are used to create the synthetic data are also used to recover the unknown fluxes.

20. p. 9, section 2.5.4: An enhanced estimation of all parts of total uncertainty is necessary to evaluate the applied range of model-data mismatch values.

The reviewer points out that in Section 2.5.4, model-data mismatch is described as including sources of uncertainty for the simulation of atmospheric $CH_4$ concentrations, which includes measurement uncertainty, transport uncertainty, and representation error. In some cases, uncertainty in the characterization of background concentrations is also part of this term. However, the evaluation of the applied range of model-data mismatch values (Sect. 4.4) previously only considered the measureable quantities of that uncertainty. We find it justifiable to report only those measures of model-data mismatch that are directly measureable in our field study. However, to satisfy the reviewer's request, we have added an additional analysis that calculates a simple approximation of the transport uncertainty for the field test that is used for evaluation. Details of the simple transport uncertainty calculation are found in the Supplemental Information, and a description of the additional uncertainty this might be expected to add, along with clarifications to the existing text, have been included in Section 4.4 (page 18, lines 5-15).

21. p. 10, l. 9: Please give an example for 'long periods of time'.

The sentence in Section 2.6.1 has been edited for clarity (page 12, line 7). The implementation of the described system characteristics enable it to run autonomously for any period of time.

22. p. 11, l. 1: It can be expected that a local wind circulation is developing at Table Mountain. The assumption of homogeneity and stationarity of turbulent fields is highly questionable under these conditions (Gaussian plume model for atmospheric transport). It has to be checked that the assumptions of the Gaussian plume model are valid for the investigated location and time period.

The reviewer points out that meteorological conditions are not homogeneous or stationary through space. Measuring the entire wind field would, of course, be ideal for modeling of the environment, but the number of sensors required would not be practical. Following previous work (e.g., Hirst et al., 2004), we generalize the wind field from measurements made at a nearby location with a sonic anemometer. We have added an acknowledgement of this generalization, and the possibility that the anemometer measurements do not perfectly represent the conditions influencing the plume, to Section 2.6.4 (page 13, lines 4-7). Additionally, an estimate of transport uncertainty that includes uncertainty in the position of the wind sensor (which is assessed by examining a timeseries of data taken simultaneously by two anemometers located at different locations on the Table Mountain field site), has been added to Section 4.4 (page 18, lines 5-15) and in greater detail to the Supporting Information.

The reviewer also discusses the applicability of the Gaussian plume model for turbulent fields that are not homogeneous or stationary. The plume model endeavors to represent the mean state of fluid flow in the atmosphere, averaged over time and space, given conditions of relatively constant wind direction and speed through time and space (absolute constancy of conditions are, of course, not realistic). By using atmospheric measurements that are averaged over time and horizontally through the plume, and at a flat terrain location with no physical or topographic obstructions to flow, we find that the application of the plume model meets the basic assumptions of validity. Further, we use a short window of averaging time (2 minutes) to ensure that wind direction and wind speed conditions do not change. Finally, the Gaussian plume model requires conditions of a constant, mass-conserving pollutant source, both of which are met in these tests. In this sense, the assumptions of the Gaussian plume model are valid for the investigated location and time period. The authors agree that more sophisticated representations of atmospheric flow could be helpful for future studies focusing on source estimation in field settings. In Sections 2.1 (page 5, lines 17-20) and 6 (page 20, lines 20-31) we have added new discussion on this topic. In particular, we highlight that, while more sophisticated models are better adapted for realistic representation of atmospheric flow, there is value in applying a "Jane Doe" model that is accessible and well-known by the general scientific community. It is assumed that the use of better models would likely lead to better resulting flux estimation with the field data presented here. Using a simple model provides a baseline for performance, however, and demonstrates, at the most basic level, the potential of the proposed methodology for methane leak detection.

23. p. 11, l. 5: Can you give a reference for the applied threshold of 0.8 m/s?

We rely on the standard assumptions of Gaussian plume model application that the reliability of the method begins to decline below approximately 1 m s$^{-1}$ (e.g., De Visscher, 2013). We have added clarification of this choice and a reference to section 2.6.4 (page 13, lines 8-9).

24. p. 14, l. 2: The (short) averaging time of 1-2 min disagrees probably with the assumption of stationarity of the Gaussian plume model. It is questionable if the assumptions of atmospheric transport model are satisfied. At least, an enhanced contribution of uncertainty should be included in the analysis (and conclusions).

We refer the reviewer to the edits to the manuscript in response to "Specific Comment" number 11, above, in response to the comments here regarding the applicability of a steady-state model over a 120-second averaging time, and the assumptions of the atmospheric transport model used. In summary of that point, it has been shown that when measurement averaging time is comparable to transport time between source and measurement point, a steady-state model such as the Gaussian plume model is appropriate (Gifford, 1976). For added clarity on this subject, we have edited the text in Section 2.1 (page 5, lines 27-30).

The reviewer makes a helpful suggestion regarding clarification of whether or not contributions of uncertainty from the choice of or assumptions associated with the transport model are included in the analysis. With respect to inclusion of enhanced contribution of uncertainty (due to choice of transport model) in the analysis:

In the synthetic data tests, the levels of "noise" added to the data are not meant to represent quantifiable combinations of the various sources of model-data mismatch uncertainty. That is, added noise of 1 ppb is not, for example, broken down into the components of uncertainty contributing to that overall value. In that sense, contributions of transport uncertainty are indeed included in the synthetic tests.

In the field data tests, so-called "bottom-up" estimations of observation uncertainty are not part of the analysis, but uncertainty due to transport modeling is, again, an important and included part of the analysis. The NZMB method first uses an NNLS approach to fitting a flux value to the observations, for which estimation of "bottom-up" observation (i.e., model-data mismatch) uncertainty is not necessary. In the second step, the NZMB method uses the actual residuals of the fit to the data as a measure of uncertainty in that term. This means that contributions of transport uncertainty are included in the observation (i.e., model-data mismatch) uncertainty.

We address the need for clarification of this point with added text to Section 2.4 (page 7, lines 24-27), in response to "Specific Comment" number 16, above.

In response to the reviewer's suggestion that a discussion of transport model contributions to uncertainty be added to the Conclusions, we point to added text in Section 6 of the paper that addresses this point.

25. p. 16, section 6: The critical discussion of the used atmospheric transport model is missing here. If a simple assumption (Gaussian plume model) is used for real (complex) environmental conditions, a comprehensive analysis of uncertainties is needed to evaluate the potential and the general applicability of the method.

We have added substantial discussion to the paragraph cited here at the reviewer's request (page 20, lines 20-31). The simplicity of the Gaussian plume model in achieving the correct identification of leaks in a field test serves here to demonstrate the viability of the measurement system and statistical technique for leak detection. Included in the added text is discussion of the need for a future analysis of transport models to identify which would be most suitable for the identification of leaks using the DCS system and NZMB method. This comprehensive analysis

would be best suited for a future study of that problem, as to undertake such an analysis would not be directly related to the central themes and goals of the work in discussion here.

26. Fig. 10: Are the concentration data referred to the length of the beam? Please provide the wind direction according to meteorological conventions: 0 deg = north, 180 deg = south, 90 deg = east, 270 deg = west). Which averaging time was used to provide the data?

The concentration data are corrected for beam length. The data shown therefore depict line-averaged (or line-integrated) concentrations, as described in the text.

The caption for Figure 10 has been edited to specify wind direction according to meteorological conventions, as recommended by the reviewer.

The averaging time used to provide the data is 2-minutes. This has been clarified in the caption for Figure 10.

Technical corrections
1. p. 3, l. 4: 'Coburn' – 'et al.' is missing.

This typo has been corrected.

2. p. 5, l. 28: 'Truong et al.; Waxman et al.' – The year is missing.

This typo has been corrected.

3. p. 5, l. 32: 'Coburn' – 'et al.' is missing.

This typo has been corrected.

4. p. 5, l. 33: 'tempus lorem' – misprint?

This typo has been corrected.

5. p. 8, l. 8: 'that that' – delete one 'that' 6. Table 2: Source Location 1 and NZMB solution: unit must be kg/s

This typo has been corrected.
* * *
Anonymous Referee #2:

In atmospheric "inverse dispersion" problems, we attempt to deduce something about gas transfer to or from the atmosphere by (i) sampling the gas concentration field, (ii) simultaneously gathering sufficient meteorological data to characterize atmospheric transport (mean wind and turbulence), then (iii) invoking an atmospheric dispersion model to make inferences about the source(s). Within that overall scheme there are many variants. Such problems arise and can be solved on every meteorological scale of atmospheric (or, for that matter, oceanic) motion. Sometimes one knows the exact location of the source(s) and the object is their quantification; sometimes deducing source location (in space and or time) is the most important aim; and sometimes one seeks to deduce source space-time location *and* strength.

This paper is highly relevant to the problem of determining by inverse dispersion the spatial location(s) of point- or near-point sources in the context of gas leak detection (specifically methane) from industrial sites, and an element that is stressed is the avoidance of false positives (for potential leak locations) while not failing to find actual (non-zero) source locations. The authors briefly describe what appears to be a highly capable and useful instrument for their purpose (though describing the instrument is not the key aspect of this paper), but mostly focus on their statistical strategy for optimizing the useful information that can be extracted from their "measurements" (the quotes, because the paper invokes measurements both synthetic and real). I have not been able to fully comprehend that statistical strategy, and I feel the recipe for it can and should be clearer. Some readers may wonder why a Bayesian framework has not been adopted, as perhaps the most rational way to make use of prior information and account for uncertainties.

We thank Referee 2 for the time that they have taken in carefully reviewing this work and for their helpful comments.

We have edited the text in Section 2.4 (page 7, line 31 – page 8, line 13) to clarify the recipe for the statistical strategy. The changes to the text include expanded descriptions of the methodology and simplifications of the nomenclature.

The reviewer raises an important question regarding why a Bayesian framework was not adopted. Given the lack of clarity in the previous draft of the manuscript regarding the dimensions of *m* and *n* (which we have tried to address in the new draft, particularly in response to the Specific Comments, below), we see why the reviewer would have suggested a Bayesian approach. Given that *n* is greater than *m* in the flux estimation problems presented here, we find that our approach is suitable, compared with approaches (such as the Bayesian approach) that are typically used for ill-conditioned (under-determined) problems, or problems in which the number of data points used to inform the fluxes are equal to or fewer than the number of fluxes to be resolved.

In this manuscript, we describe a leak detection strategy with the dual goals of identifying and quantifying potential leaks while limiting the instances of false positive leak detection. The

proposed method shows promise to address both goals with accuracy.

Setting aside that aspect of the paper, to my mind the most vulnerable aspect of the authors' methodology is their use of the Gaussian plume model as their atmospheric transport paradigm, which treats the turbulent surface layer wind as if it were a regime of unsheared homogeneous turbulence. The authors do recognise that the Gaussian plume model is highly simplistic, and I can accept their argument that its use in the context of their paper is acceptable. But to drive home the importance of the choice of wind model, I must stress that the mean wind speed and the effective eddy difffusivity vary radically with height across the atmospheric surface layer (ASL), in a manner that is well described by Monin-Obukhov similarity theory. There are much higher fidelity models available, and one of these could be substituted without great penalty in terms of computational burden. It is true that at some sites, obstacles or topography may disturb the transporting wind field such that it is more complex than envisaged even by the better models (some of which are listed below), but they cannot be a worse choice than the Gaussian plume model, which is in effect a mental straitjacket.

We recognize the reviewer's concerns about our use of the Gaussian plume model, and address this issue in our response to the related comment number 1, below.

Specific Comments
1.    The authors' admit that the Gaussian plume model (GPM) is highly simplistic, and I accept their argument that its use in the context of their paper is acceptable. I would disagree with their *categorical* (i.e. no exceptions) assertion that the GPM is "more suitable" if the inversion is based on line-averaged concentration data. The GPM is unnecessarily simple. There are much better analytical solutions that could and should be used to evaluate the H matrix. Whereas the GPM entirely neglects mean wind shear and treats the eddy diffusivity as constant (the atmospheric surface layer being represented as a regime of unsheared homogeneous turbulence), better models represent the ASL with a mean wind shear and a height-dependent eddy diffusivity that are consistent with the state of the ASL as parameterised by the Obukhov length and friction velocity. These solutions can be rapidly calculated; and where they provide (only) cross-wind integrated concentrations, the authors could easily introduce Gaussian *crosswind* spread (note: it is hard to improve on the assumption of Gaussian crosswind spread without using measured data on wind direction fluctuations).   A better analytical (or semi-empirical) eddy diffusion model could be sought out from the following references: Philip (1959), Ermak (1977), van Ulden (1978), Nieuwstadt and van Ulden (1978), Huang (1979), Lupini et al. (1981), Wilson (1982), Kormann and Meixner (2001), Sharan & Kumar (2009), Wilson (2015).

The reviewer is correct; the sentence the reviewer refers to ("more suitable") was poorly phrased and led to the implication that the Gaussian plume model is more suitable than other models in this context, which is not the case. We emphasize here and with additional text in the manuscript (see Section 2.1 and Conclusions) that, while more sophisticated models are better adapted for realistic representation of atmospheric flow, there is value in applying a "Jane Doe" model that is accessible and well-known by the general scientific community. It is assumed that the use of better models would lead to better resulting flux estimation with the field data presented here. Using a simple model, however, provides a baseline for performance and demonstrates, at the most basic level, the potential of the proposed methodology for methane leak detection.

We have replaced the text referred to by the reviewer (Section 2.1, page 5, lines 15-24) with a note on the drawbacks of the plume model, and recommendations that future studies focused on field data or applications of the methodology for leak detection purposes consider employing a more sophisticated plume model such as those suggested, AERMOD or a stochastic Lagrangian particle model. Additionally, we have included additional discussion – in the Conclusions section of the manuscript (page 20, lines 20-31) – that more realistic representation of atmospheric transport could be achieved with the use of alternative models.

2.      p5, line 30: Is there an easy argument that the optical detector's response is to line-averaged methane mole fraction as opposed to line averaged mass concentration [kg m$^{-3}$]?

The reviewer is correct on this matter; the absorption measurement is proportional to the concentration (number density) of the absorbing gas along the line of site (laser beam) rather than directly related to the mole fraction - which is calculated using the temperature (derived from the absorption spectrum) and pressure (from an external pressure monitor).

We have adjusted the text to reflect the reviewer's important clarification on this matter (page 6, line 21).

3.      At p6 line 5, why "attempts to solve"?

The reviewer is correct. The NNLS algorithm does not "attempt to solve" the least squares problem; it computes value(s) of $x$ that solve the least squares problem. We have edited the text highlighted by the reviewer for accuracy in this regard (page 7, line 1).

4.      I don't understand why the authors contend (p6, line 19) that (in general, with m source locations and n concentrations) their "problem is overdetermined" — do they assume n > m?

The dimensions $n$ and $m$ were previously not clearly defined, and the reviewer has helpfully identified this problem in both this comment and comment number 7, below. In fact, the dimensions of $n$ are greater than the dimensions of $m$, such that the problem is overdetermined. Multiple measurements along each beam are used in each fit, so that $m$ is never less than $n$.

To clarify this important point, we have adjusted the text in the following sections:

In Section 2.1 (page 6, lines 10-11), we have added the sentence: "In the synthetic tests and field tests described here, multiple measurements are made along each beam, such that $n$ is always greater than $m$."

In Section 2.4 (page 7, line 22), in the sentence identified by the reviewer in this comment, we have added the clarification: "the problem is overdetermined (that is, $n > m$)."

In Section 2.5.1 (page 9, lines 1718), the value of $m$ for each test is explicitly defined ($m = 20$).

In Section 2.5.3 (page 10, lines 29-32), we have added the clarifying sentences: "In the synthetic tests, the dimensions of $n$ (e.g., the length of the atmospheric concentration vector, $\mathbf{c}$) vary along with the number of beams per spectrometer-detector system and the number of meteorological conditions. In the configuration of 4 beams, for example, $n = 216 * 4$, because each distinct meteorological condition is applied to each beam. In the 8 beam configuration $n = 216 * 8$, in the 16 beam configuration $n = 216 * 16$, and so on." A reference to this Section has also been added to Section 2.5.4 (page 11, line 21).

In Section 2.6.2 (page 12, lines 21-23), the dimensions of $m$ are specified for each case (in both cases, $m = 1$).

In Section 2.6.5 (page 13, lines 23-29), a new paragraph has been added that explicitly names the values of $n$ for both field test cases: that for source location 1 and that for source location 2. For source location 1, all downwind measurements along beams (retros) 1 and 2 are used in the least squares fit, such that $n = 63$. For source location 2, all downwind measurements along beams (retros) 2 and 3 are used, such that $n = 30$.

5.  Why (p6 lines 22-23) should it be the case that (or why is it a safe assumption that) "model-data mismatch uncertainty has an un-biased Gaussian distribution"?

    The reviewer is correct in pointing out that it is not a safe assumption to suppose that there is no bias in the distribution of the model-data mismatch uncertainty. We have amended the text to reflect this point (Section 2.4, page 7, lines 28-29). For the purposes of the methods and analysis here, we keep the un-biased model-data mismatch because we believe it will take a separate and extensive study to examine realistic distributions of uncertainty for the test cases posed. This assumption follows on previous observing system simulation experiments and other work focused on emissions detection. For example, from Crenna et al. (2008): "Model error also arises unavoidably because numerical models are based on idealized relationships that are only an approximation of the real world. Such idealizations presumably introduce systematic errors in $a_{ij}$, but it is outside the scope of this paper to attempt to address such errors."

6.  I find the derivation (p6 line 31 to p7 line 2) hard to follow, yet I suspect it has to be very simple. For instance we have the three symbols $\varepsilon_{Ri}$, $\varepsilon_R$ and $\varepsilon_{bi}$: is the last just the first, in alternative guise?

    The reviewer is correct that $\varepsilon_{Ri}$ and $\varepsilon_R$ are the same (the latter is simply in vector notation), and $\varepsilon_{bi}$ is similar: it represents a bootstrapped sampling of $\varepsilon_R$. That is, $\varepsilon_{bi}$ is a permutation or resampling of $\varepsilon_{Ri}$. In the text, we have removed the subscript R and replaced $\varepsilon$ with e, which is a more correct notation, so that the two vectors are now simply $\mathbf{e}$ and $\mathbf{e}_b$, to make it clearer to the reader that the latter is a bootstrap sampling of the former, not a new value altogether (page 7, line 31 – page 8, line 13).

7.  If I have understood correctly, the pool of residual values is a set containing only n members, where n is the number of concentration measurements (n = 3 for the field test). Then, for each detector (i = 1...n) one randomly draws 1000 samples from that set, with replacement, thereby

obtaining a set of 1000 alternative model predictions $y_{bi}$ for (each) source location. The logic for this is not very sound, it seems to me, because all observations are given equal status, irrespectively of their distance from the source(s). In real world cases there could be order-of-magnitude differences in the measured mean concentrations — and indeed in concentration variance and higher moments that, although irrelevant here, surely relate to the trustworthiness or representativeness of a measurement — and in the level of uncertainty in the modelling.

Firstly, we thank the reviewer for pointing out that the values of $n$ (and indeed $m$) require clarification in the manuscript. The value of $n$ is substantially larger than 3 in the field tests ($n$ is 30 or more) and in the synthetic tests ($n = 864$ or more). We refer to the text changes documented in response to comment number 4 for clarity on this matter.

The reviewer brings up a second important point regarding the treatment of all observations as having "equal status". For the field tests, it becomes evident through the clarifications outlined in response to comment number 4 that the observations used in the fit (and therefore in the bootstrap) occur on the same beam, if (as is the case in the field data presented here) the wind direction is such that the downwind direction does not change throughout the course of sampling. For that reason, different observation residuals resampled in the bootstrap do have "equal status" in terms of the distance between source and receptor.

However, a corollary to the reviewer's point is that, even in the field test cases (in which the distance to the downwind beam does remain static throughout the experiment) changes throughout the measurement period in the meteorological conditions may mean that not all observations should be treated equally in the bootstrap analysis. For example, differences in the measured concentrations due to changes in wind speed, wind direction, and atmospheric stability could be expected to result in changes in the measurement representativeness and uncertainty in modeling.

To address this issue, we have adjusted our analysis to now implement a moving block bootstrap (replacing the bootstrap used in the NZMB method). The moving block bootstrap method recognizes the short-range dependence of measurements collected under similar conditions and, relatedly, the independence of measurements not collected under similar conditions. We refer the reviewer to Künsch (1989) for a full description of the moving block bootstrap technique. We calculate the autocorrelation of the residuals to determine the length of time appropriate for the moving block bootstrap, or the time "window" over which residuals are resampled for a given observation. We take two times the length of time at which residuals demonstrate autocorrelation at the 95% confidence level. The autocorrelation length is doubled to ensure that a sufficient number of observations are included in the moving window to provide a robust statistical sample size. For the measurements at source location 1, the moving block window is 78 minutes in length and for the measurements at source location 2, the moving block window is 114 minutes in length.

The same change in methodology is not applied to the synthetic tests because there is no time dimension over which to apply a moving block bootstrap. Field tests or synthetic tests using time-resolved meteorological data could use the moving block bootstrap, and the block bootstrap

could further be restricted to sampling of residuals on a beam-by-beam basis for source-receptor configurations that more closely mirror those in the synthetic tests.

Text has been added to section 2.4 (page 8, lines 1520) to describe and reflect the methodological updates described here.

8.     At p7, line 13, it might be helpful to be more specific as to what "law of large numbers" means in this context. I expect it amounts to an assumption that the distribution of some mean value (or sum) is Gaussian, even if the numbers being summed do not have a Gaussian distribution (central limit theorem)?

We agree that clarification on this matter is needed, and have added text to Section 2.4 (page 8, line 32 – page 9, line 3) to that end. We rely on statistics literature, which states that, even for the dependent case, under certain regularity conditions on random variable moments, the sample mean of the random variables approaches the population mean. This holds true for the bootstrapped case as well. That is, when the number of bootstrap operations is large, and given the sample, the bootstrapped leak mean approaches the estimated leak from the sample ("law of large numbers"). The Central Limit Theorem (CLT) states that under appropriate scaling for large sample size, the test-statistic (or estimated leak rate) converges in distribution to a normal distribution. When the number of bootstrap operations is sufficiently large and $n$ is also large, then, assuming the residuals are independent, it has been shown (Bickel and Freedman, 1981; Singh, 1981) that the bootstrapped leak mean admits a CLT. That is, the bootstrapped mean converges in distribution to a normal distribution with mean being the unknown population leak (or leak to be estimated) and scaling factor as the square root of $n$.

9.     At p9 line 21, the authors allude to "model-data mismatch noise." It seems to me that "noise" type errors (which can be dealt with by averaging) are in practice likely to be far less serious than *systematic* errors arising from the imperfect modelling of atmospheric transport.

The reviewer is correct that the word "noise" evokes the kind of uncertainties that can be easily averaged out, whereas model-data mismatch uncertainties may not always be Gaussian. In the paragraph in question, we use the word "noise" rather than the word "uncertainty" to emphasize that we perturbed the synthetic data to create an observing system simulation experiment (OSSE). We do, however, change several instances of the word "noise" to "uncertainty", as the discussion in the first part of the paragraph does focus on real uncertainties associated with transport and measurement precision. The title of section 2.5.4 (page 11, line 1) has been change to reflect the reviewer's point, and text edits have been made throughout that section (particularly page 11, line 15-20).

A related point that the reviewer makes is that Gaussian noise is potentially less problematic for inversions than would be systematic errors or uncertainties in the ability to simulate observations. Text added to Section 2.4 (page 7, lines 28-29) acknowledges this possibility. For the purposes of the study presented here, we apply the traditional OSSE framework of perturbing the observations with Gaussian noise. Ideally, measurements would be gathered for a long enough period of time so that transport and other errors would assume a Gaussian distribution, however this is not always possible in limited-scale field studies and may not always be the case

regardless of the length of the dataset, given dispersion model uncertainties. Future studies of systematic biases in model-data mismatch and impacts on flux estimation would therefore be warranted.

10.     How is gas-gas interference dealt with? What about detection of "stray" infra-red radiation emitted from the environment and/or from within the telescope?

Gas-gas interference is avoided in dual-comb spectrometry due to the large spectral bandwidth (~60 nm) and simultaneous high spectral resolution (0.002 nm). The bandwidth and resolution allow the system to fit and distinguish the individual absorption features and patterns for each gas - even when the patterns overlap.

"Stray" or background infrared light does not affect the laser signal measured on our fast photodiode detector due to the heterodyne nature of the detection. "Stray" signals do appear at low frequencies, but these signals do not affect the high frequency heterodyne beat signals between the comb teeth in dual-comb spectrometry.

We have added clarification of these points in Section 2.6.1 (page 11, line 31 – page 12, line 4) to help readers who will have similar questions.

11.     Inversion of the H matrix can entail severe error if the matrix is "ill-conditioned" as a result of the relative positioning of the sources and detectors: for example if sources are aligned along the wind direction (see Crenna et al. 2008; Flesch et al. 2003).

The reviewer raises an important point and suggests several seminal papers on the topic of sensor placement for optimization of the condition number of the H matrix. In the experiments presented here, we sought to test a case study that was agnostic to sensor placement, as that topic merits a full study of its own for the particular distribution of point sources such as is posed by oil and gas operations. By randomly distributing beams among wells, assuming known background concentrations, and incrementally increasing beam density in a way that is agnostic to point source location, we hope to provide the simplest test of the NZMB method. This point is made in the study's Conclusions Section 6, but, as the reviewer points out, it needs to be emphasized. We have added text to that section of the paper citing the papers referenced by the reviewer here as examples of studies focused on the benefits (and drawbacks, if not properly considered!) that sensor placement can pose (page 20, lines 13-17).

In order to create a field experiment that closely matched the synthetic experiment condition of perfectly-known background concentrations, we relied on sensor placement such that the background concentration was measured independently from signals arising from local sources, such as Crenna et al. (2008) suggest. Similarly, sensors are placed so that a given sensor is unlikely to measure emissions from more than one source, achieving "two nearly independent single-source problems", as also recommended by Crenna et al. (2008). These beam placements serve to minimize $k$, and will certainly inform future studies aimed at sensor placement in complex or dense fields of wells and facilities.

Finally, as Crenna et al. (2008) points out, more measurements (particularly under different

sampling conditions) can lead to lower condition numbers, even in cases of sub-optimal sensor placement. The synthetic experiments here are designed to capitalize on this characteristic, by simulating a wide range of wind conditions. Field experiments and future field sampling of emissions from real oil and gas facilities will similarly benefit from longer measurement timeseries to maximize the number of independent tracer interactions with sensors.

To address these points, the text has been edited and appropriate references included in Section 2.5.2 (page 9, lines 24-26 and page 10, lines 1-3), Section 2.6.2 (page 12, lines 17-19), and Section 6 (page 20, lines 13-17).

12. Using the GPM entails selection of appropriate σ-curves: the choice should be documented.

The reviewer is correct – thank you for identifying this oversight. We have added a description in Section 4.2 (page 16, lines 26-29) of the stability classes and σ-curves used for the field data, and have added more detailed description of the calculation of the σ-curves in the Supplemental Information.

13. The Conclusion does not, to my mind, sufficiently recognise the potential accuracy gain from using a more sophisticated and realistic atmospheric transport model.

The co-authors agree that this issue needs to be revisited again in the concluding section of the paper. A paragraph addressing the use of the GPM (and recommendations to use more realistic simulations of transport) has been added to the final (Conclusions) Section (page 20, lines 20-31).

14. I wondered whether the authors are familiar with normal practice in regard to averaging, in micrometeorology. To make sense of surface layer winds of course we necessarily must use statistics, and it is usual to base those on an averaging interval of at least about 15 minutes. However the discussion caused me to suspect that the authors envisage using nearly-instantaneous wind measurements (or, say, 1 min averages) to deduce leak rates from concentration averages over (say) one or two minutes. This may be feasible, indeed it may be a good idea (at the least, it would speed up a search across many potential leak sites). The thing is, however, that one has to recognize that existing, documented, tested, trusted surface layer dispersion models exploit a statistically stable estimate of surface layer state, and that statistical stability demands averaging intervals much longer than a minute (or two). To adapt existing atmospheric transport modes to very sort averaging intervals one will at the least have to adjust the parameterisation of the eddy diffusivity.

The reviewer makes a valid point that it is potentially problematic to apply parameterizations of atmospheric dispersion to 2-minute datasets that were developed for longer time scales, and generally to interpret dispersion models at short timescales. The reviewer aptly identifies that it is valuable for the purposes of the application presented here to obtain information over short time frames, in particular to speed up the search across many possible leak locations. Further, this approach is necessary because background methane concentrations fluctuate with extremely high frequency (order minutes) in proximity to natural gas operations. Observational datasets

that would average across these fluctuations would likely miss the small signals that we attempt to recover here.

While this issue may remain a substantial and unresolved caveat of this methodology, we point to several characteristics of the observing framework that may help to mitigate its impact. We also perform a sensitivity test to analyze the impact of adjustments to the eddy diffusivity parameterizations.

Emissions from oil and gas are typically close to the ground surface (natural gas wells are not generally taller than several meters in height). Our sensors are typically also located near the surface. For this reason, vertical fluctuations in the plume between the source and receptor may be expected to be truncated by interaction with the ground surface, which, in theory, would mean that sensitivity to shorter time period oscillations should be damped in comparison with source-sensor configurations located higher above the ground level. Weil et al. (2012) demonstrate this idea, for example, by showing that crosswind-integrated concentration fluctuation intensity is smaller for sources closer to the surface than for elevated sources. This provides at least some assurance that shorter averaging times may be less problematic at the surface than higher above ground level.

Despite the US EPA's recommendation that hour-long averaging times be used for application of the Briggs dispersion parameterizations, the Briggs parameterizations were derived using 10-minute averaged data (Beychok, 2005). The Australian EPA recommends use over 3-minute time frames. It is possible, therefore, that the discrepancy between the time averaging used in the work presented here is not as vast as, for example, the EPA recommendations would imply. In recognition of the discrepancy between the averaging time for which the Briggs parameterizations were developed (10 minutes) and the averaging time used here (2 minutes), we calculate the expected potential error introduced in sigma-y using the standard equation for the adjustment of dispersion parameters for different averaging times:

$$\sigma_{y,2} = \sigma_{y,1} \left( \frac{t_2}{t_1} \right)^p$$

This equation can be found in Gifford (1976).

If the Briggs parameters used here were calculated using an averaging time of 10 minutes, then $t_1$ = 10 minutes. We use measurements averaged over 2 minutes and meteorological data averaged over 2 minutes, so that $t_2$ = 2 minutes. Using the typical value of 0.2 for the empirical parameter $p$ (for $t_2 < 1$ hour), we anticipate a factor of 0.7 correction to the Briggs dispersion parameter $\sigma_y$.

Using the above recommended method for adjustment of $\sigma_y$, we re-calculate the results shown in Table 2 as a sensitivity test. We are not aware of an equivalent equation for the adjustment of $\sigma_z$. We find that the sensitivity of our result to the adjustment to $\sigma_y$ is negligible: the value of the non-bootstrap solution for Source Location 2 changes by 0.1 E-5 kg s$^{-1}$. This is, perhaps, to be expected, given that the cross-wind fluctuations are largely averaged through by the laser beam when winds are orthogonal to it. We report this sensitivity and show histograms of the results from this sensitivity test in the Supplemental Information section.

| | Source Location 1 | Source Location 2 |
|---|---|---|
| **Controlled Leak Time On:** | 10:08 | NA |
| **Controlled Leak Time Off:** | 16:30 | NA |
| **Measured Mean Flow Rate:** | 3.1 E-5 ± 0.01E-5 kg s$^{-1}$ | 0.0 ± 0.0 kg s$^{-1}$ |
| **Non-Bootstrap Solution:** | 2.4 E-5 kg s$^{-1}$ | 0.5 E-5 kg s$^{-1}$ |
| **NZMB Solution:** | 2.6 E-5 ± 0.5 E-5 k/s | 0.0 ± 0.0 kg s$^{-1}$ |

A final note on averaging time, and perhaps the most relevant defense of the methods we lay out here, is that the separate dispersion calculations (which are based on 2-minute averages) are aggregated together in a single least-squares fitting routine that spans much longer time frames (several hours). This means that unresolved or unresolvable errors in the vertical and horizontal dispersion over each 2-minute interval may be resolved, in the mean, over longer time periods. This approach is similar to that taken in CALPUFF, for example: when averaging times exceed 1 hour, the algorithm applies averaging of each separate 1-hour dispersion calculation (Scire et al., 2000).

We find, for the goals of the methods developed here, that the losses in model fidelity incurred by averaging over short timescales (2 minutes) are acceptable given the particular requirements of methane leak detection (resilience to rapid background variability, rapid scanning for potential leaks). This is particularly true given that 2-minute intervals are aggregated over longer time periods to obtain solutions. The promising results shown here suggest that this trade-off, of more accurate parameterizations for a useful methane leak detection solution, might be worth making.

We have included a discussion of this important topic in Section 5.2 (page 19, line 22 – page 20, line 3) of the paper.

Technical corrections
1. First paragraph of Sec. 2.6.2 uses mixed tenses (present then past).
   The tenses have now been made the same in this section.

2. Figure 3 shows only two beams for the field test, whereas the discussion of the experiment alludes to three.
   Thank you for catching this mistake. We have replaced Figure 3 with the correct version, which shows all three beams and both potential leak locations.

3. The number of panels on Figures (5,6) seems excessive — is it necessary to cover the range in MDM (model-data-mismatch) with such fine steps.
   Thanks to the reviewer for this suggestion. We have adjusted Figures 5 and 6 to only show a subset of the MDM results: 0.5 ppb, 1.0 ppb, 2.5 ppb, 5.0 ppb, and 10.0 ppb. Accordingly, the text in the paragraph beginning on page 15, lines 5-7 has been adjusted. Language was also added in Section 3.1. to guide the reader in the interpretation of these figures, as they are complex. An issue with figure numbering has also been fixed. We hope that these adjustments improve readability. We have included the full range of MDM in the Supplemental Information.

4. The structuring of this paper results in a degree of repetition.

We removed repetitive phrases, sentences or paragraphs from the following sections: 1, 2.1, 2.4, 2.5.2, 2.5.3, 3.2, 4.3

[Figure]

**Figure S1: Timeseries of stability class calculated for Table Mountain field site on 01/26/2017.**

As a sensitivity test, we examine the use of dispersion coefficients that are based on 10-minute empirical data for our 2-minute datasets. We calculate the expected potential error introduced in $\sigma_y$ using the standard equation for the adjustment of dispersion parameters for different averaging times:

$$\sigma_{y,2} = \sigma_{y,1} \left(\frac{t_2}{t_1}\right)^p, \tag{S1}$$

found in Gifford (1976).

The Briggs parameters used in this study were calculated using an averaging time of 10 minutes, so that $t_1$ = 10 minutes (Briggs, 1974; Griffiths, 1994). We use measurements averaged over 2 minutes and meteorological data averaged over 2 minutes, so that $t_2$ = 2 minutes. Using the typical value of 0.2 for the empirical parameter $p$ (for $t_2$ < 1 hour) (Gifford, 1976), we calculate that a factor of 0.7 correction should be applied to the Briggs horizontal dispersion coefficient, $\sigma_y$. We are not aware of similar adjustment formulas for the vertical dispersion coefficient.

We apply the correction factor to the field data collected on January 26$^{th}$, 2017 (the synthetic data has no time dimension, and transport is considered perfect in that observing system simulation experiment, so the choice of the dispersion coefficient has less bearing on the findings). The results are seen in Table S1.

| | Source Location 1 | Source Location 2 |
|---|:---:|:---:|
| **Controlled Leak Time On:** | 10:08 | NA |
| **Controlled Leak Time Off:** | 16:30 | NA |
| **Measured Mean Flow Rate:** | 3.1 E-5 ± 0.01E-5 kg s$^{-1}$ | 0.0 ± 0.0 kg s$^{-1}$ |
| **Non-Bootstrap Solution:** | 2.4 E-5 kg s$^{-1}$ | 0.5 E-5 kg s$^{-1}$ |
| **NZMB Solution:** | 2.6 E-5 ± 0.5 E-5 k/s | 0.0 ± 0.0 kg s$^{-1}$ |

**Table S1: Same as Table 2 in the main text, but calculated with an adjustment to the dispersion coefficient, $\sigma_y$. Controlled methane release flow rates and 1 standard deviation for each field experiment, including local time that leak was turned on and off.**

The recovered fluxes do not change to within a detectable range. It is possible that the sensitivity of our result to the adjustment to $\sigma_y$ is low because the cross-wind fluctuations are largely averaged through by the laser beam when winds are orthogonal to the beam. The histograms of the solutions shown in Table S1 are shown below.

[Figure]

**Figure S2: Same as Figure 11 in the main text, but calculated with an adjustment to the dispersion coefficient, σy. Histogram of NZMB estimated source strength at source location 1, with dashed line showing the bootstrap mean and thin dotted lines showing ± 2 standard deviation. The thick black line shows the true leak strength at source location 1 (3.1 E-5 kg s$^{-1}$).**

[Figure]

5    **Figure S3: Same as Figure 12 in the main text, but calculated with an adjustment to the dispersion coefficient, σy. Histogram of NZMB estimated source strength at source location 2, with dashed line showing the bootstrap mean and thin dotted lines showing ± 2 standard deviation. The thick black line shows the true leak strength at source location 2 (0 kg s$^{-1}$). The presence of 0 kg s$^{-1}$ in the histogram triggers acceptance of the null hypothesis (that the emissions rate at this site is zero).**

10   **Full range of Model-Data Mismatch Results**

Here we show the full range of model-data mismatch results, which are not shown for the sake of avoiding figure complexity in Figures 5 and 6 in the main manuscript.

[Figure]

**Figure S4: Same as main text Figure 5, but with all model-data mismatch cases shown.**

[Figure]

**Figure S5: Same as main text Figure 6, but with all model-data mismatch cases shown.**

**Estimation of transport uncertainty**

5     A simple estimate of transport uncertainty is derived by considering the combined impacts on simulated atmospheric concentrations of meteorological measurement instrument uncertainties, parameterization of atmospheric stability, and placement of the sonic anemometer relative to leak location (that is, the influence of using a point measurement of wind speed to characterize the entire wind field over the Table Mountain area). We use meteorological measurements made by two separate sonic anemometers placed at different locations (several hundred meters apart) on the Table Mountain site over

10     the course of nearly 6 hours on March 3, 2017. Wind speed and wind direction are measured by both instruments. These meteorological measurements are used in a plume model simulation of the enhancement due to a leak with a rate equal to the controlled release of methane at source location 1 (3.1E-5 kg s$^{-1}$) along a 585 m beam that is either 9 m (the mean minimum lateral distance between each leak and adjacent beam in our test configuration) or 209 m (the mean distance between each leak and each segment of the beams directly monitoring it) downwind of the leak and perpendicular to wind direction. We

15     calculate the standard deviation in the enhancement along the beam given the simulations with data from different

anemometers and given an increase and a decrease in the stability of one stability class from the mean state calculated on January 26[th], 2017 (e.g., for a daily mean stability class of B, the absolute differences between B and A and B and C are averaged). This standard deviation is then taken to be the 1-σ uncertainty due to transport reported in the comparison to model-data mismatch in Section 4.4.

**5    Note on units**

The units ppb are equal to the SI units nmol mol$^{-1}$.

---

## Author Response (AR2)

[revised manuscript text omitted]

Spectrometer